# Unraveling chirality transfer mechanism by structural isomer-derived hydrogen bonding interaction in 2D chiral perovskite

Jaehyun Son[1,4], Sunihl Ma[1,3,4], Young-Kwang Jung [1], Jeiwan Tan [1], Gyumin Jang [1], Hyungsoo Lee[1], Chan Uk Lee[1], Junwoo Lee[1], Subin Moon[1], Wooyong Jeong[1], Aron Walsh [2] & Jooho Moon [1]✉

In principle, the induced chirality of hybrid perovskites results from symmetry-breaking within inorganic frameworks. However, the detailed mechanism behind the chirality transfer remains unknown due to the lack of systematic studies. Here, using the structural isomer with different functional group location, we deduce the effect of hydrogen-bonding interaction between two building blocks on the degree of chirality transfer in inorganic frameworks. The effect of asymmetric hydrogen-bonding interaction on chirality transfer was clearly demonstrated by thorough experimental analysis. Systematic studies of crystallography parameters confirm that the different asymmetric hydrogen-bonding interactions derived from different functional group location play a key role in chirality transfer phenomena and the resulting spin-related properties of chiral perovskites. The methodology to control the asymmetry of hydrogen-bonding interaction through the small structural difference of structure isomer cation can provide rational design paradigm for unprecedented spin-related properties of chiral perovskite.

Two-dimensional (2D) Ruddlesden-Popper (RP) organic–inorganic hybrid perovskites (OIHPs) have a chemical formula of $A'_2BX_4$, where A' is a bulky ammonium cation inserted between $(BX_6)^{4-}$ metal-halide inorganic frameworks. Among the many bulky ammonium cations that act as spacers to separate neighboring inorganic frameworks, bulky cation molecules can possess an inherent chirality since they cannot be superimposed onto their mirror image due to their low symmetry. Incorporating organic molecules with inherent chirality in the A'-site leads to the formation of RP OIHPs having a chiral crystal structure. Although the structural difference of RP OIHPs induced by the enantiomers is small, it can give rise to a huge difference in their spin-related properties. For example, our group initially reported that chiral RP OIHPs that utilized two different enantiomers (R- and S-methyl benzylamine (R- and S-MBA)) could exhibit circular dichroism (CD),

with an opposite sign near the first excitonic band edge depending on the handedness of the MBA cation[1].

Since the chiroptical response of chiral RP OIHPs was first reported in 2017 by our group[1], chiral RP OIHPs have been extensively investigated in chiral photonics research society. Chiral RP OIHPs are based on spontaneously self-assembled multi-quantum well (MQW) structure consisting of two alternating building blocks of metal-halide inorganic slab (wells) and bulky chiral organic spacer (barriers). Therefore, chiral RP OIHPs can exhibit both excellent optoelectronic properties derived from the inorganic slab and unique chiroptical properties induced by the bulky chiral organic spacer[2–9]. Owing to their stable and broad wavelength tunability over visible light region as well as small effective mass[10], long spin lifetime exceeding 1 ns, diffusion lengths ~85 nm, and high electron/hole mobility[11–13], chiral RP OIHPs can be an ideal candidate for optoelectronic devices based on

[1]Department of Materials Science and Engineering, Yonsei University, 50 Yonsei-ro Seodaemun-gu, Seoul 03722, Republic of Korea. [2]Department of Materials, Imperial College London, London SW7 2AZ, UK. [3]Present address: Department of Chemical Engineering, University of Michigan, Ann Arbor, MI 48109, USA. [4]These authors contributed equally: Jaehyun Son, Sunihl Ma. ✉e-mail: jmoon@yonsei.ac.kr

circularly polarized light (CPL) detection and emission. Furthermore, strong spin-orbit coupling of the electronic state[14], large Rashba-Dresselhaus splitting[15–17], and spin-dependent optical selection rules in chiral RP OIHPs suggest that chiral RP OIHPs can be also utilized into spin-polarization-based devices such as spin filter[18–21] and ferroelectrics[22,23].

However, despite of their excellent spin-related properties, the origins of the spin-polarization-based phenomena (e.g., spin-polarized photon absorption and spin-polarized photoluminescence) is still elusive. It was reported that the fascinating spin-related properties of chiral RP OIHPs are attributed to the chirality transfer phenomena from chiral organic molecules to inorganic layers[24,25]. Through symmetry-breaking lattice distortion, the inherent chirality of organic molecules can be transferred into the inorganic framework. For example, Mitzi groups have discovered that the asymmetric tilting distortion of inorganic frameworks correlates with the spin-splitting in electronic structure[24]. In addition, very recently, our group theoretically demonstrated that the asymmetric hydrogen-bonding between chiral organic molecules and inorganic frameworks can promote efficient chirality transfer phenomena by facilitating symmetry-breaking lattice distortion onto the inorganic frameworks[26]. In this respect, inducing lattice distortion of the inorganic frameworks by modulating the hydrogen-bonding interactions between the chiral organic molecules and inorganic frameworks can be recognized as one of the key factors in achieving chiral RP OIHPs with high spin-polarization sensitivity, which can elucidate the origin of spin-polarization-based phenomena.

To systematically investigate the effect of hydrogen-bonding interactions on lattice distortion and the coherent spin-related properties of chiral RP OIHPs, we need to nullify the other potential effects that might arise from the compositional change[25,27,28], molecular size variations[29], and the electronic state difference of the organic molecules[24]. However, it is hard to experimentally verify the effect of hydrogen-bonding interaction itself on spin-related properties of chiral RP OIHPs by adopting previously reported strategies. For example, when iodine anions in inorganic octahedra are replaced by bromide anions, unexpected phase transitions, and inevitable electronic structural changes occur simultaneously[29], making it difficult to focus on the effect of the hydrogen-bonding interaction (i.e., the effect of the hydrogen-bonding interactions between the halogen anion and the $NH_3^+$ amine group of chiral organic molecules on chiroptical activities in chiral RP OIHPs). To address this problem, we adopt structural isomer molecules with the same molecular formula but different functional group location as chiral spacer cation to pinpoint the genuine effect of hydrogen-bonding interactions on chirality transfer phenomena in chiral RP OIHPs by excluding the influence of compositional or electronic structural change. In this regard, the structural isomer molecule, as chiral cation spacers, can be used to gain in-depth insights into the origins of the spin-polarization-based phenomena and the chirality transfer mechanism in chiral RP OIHPs.

We selected naphthyl ethylamine (NEA) cations, which have two structural isomers of 1-(1-naphthyl)ethylamine and 1-(2-naphthyl)ethylamine (denoted as 1NEA and 2NEA, respectively) according to the position of ethylamine on the naphthyl skeleton, as chiral spacers. Because the $NH_3^+$ amine group of the chiral spacer interacts with inorganic layers via NH···X hydrogen bonding[30], the structural change of the functional group location can alter the strength of the hydrogen bonding inside the chiral RP OIHPs. Consequently, 2NEA-incorporated OIHPs show a deeper average $NH_3^+$ penetration depth toward the inorganic framework, inducing the packing conformational change of the bulky naphthyl moiety along the out-of-plane direction when compared with the 1NEA counterpart. The deeper $NH_3^+$ penetration into the 2NEA OIHPs results in a large-scale inversion asymmetric distortion of the inorganic layers, facilitating chirality transfer from the chiral molecules to the inorganic frameworks. Although our previous

result suggested that asymmetric hydrogen-bonding interaction plays a key role in modulating chiroptical activity of chiral RP OIHPs by theoretical calculation[26], the genuine effect of asymmetric hydrogen-bonding interaction on chirality transfer phenomena and spin-related properties of chiral RP OIHPs has rarely been experimentally proved. The density-functional theory (DFT) combined with experimental thin-film CD spectra and crystallographic studies verify that 2NEA OIHPs exhibit more enhanced spin-polarized photon absorption behavior than 1NEA OIHPs because of the strong hydrogen-bonding interactions between the chiral organic molecules and the inorganic framework. Our findings will contribute toward a fundamental understanding of the correlations between asymmetric hydrogen-bonding interactions and chiroptical activity.

## Results

To examine the structural isomer-derived hydrogen-bonding interaction on lattice distortion and the coherent spin-related properties of chiral RP OIHPs, we used the antisolvent vapor-assisted crystallization (AVC) method[31,32] to grow chiral (R/S-2NEA)$_2$PbBr$_4$ single crystals at room temperature (Supplementary Fig. 1), using N, N-dimethylformamide (DMF) as a good solvent and dichloromethane (DCM) as an antisolvent. The crystal structure and crystallographic parameters of each RP OIHP incorporated with structural isomers were determined by single-crystal X-ray diffraction (SC-XRD) (See Table S1 and S2 for more detail parameters). The laminar block-shaped (R/S-2NEA)$_2$PbBr$_4$ single crystal we obtained had a chiral space group of monoclinic $P2_1$, which is the same as that of the (R/S-1NEA)$_2$PbBr$_4$ single crystal[24]. For convenience, we selected the R-configuration of both NEA structural isomer molecules as a representative of each chiral RP OIHPs. It is worth noting that both isomer chiral RP OIHPs have two types of hydrogen bonding (i.e., longer hydrogen bonding (LHB) and shorter hydrogen bonding (SHB)), because of the asymmetric nature of the hydrogen bonding between the chiral organic molecules and the inorganic frameworks.

As shown in Fig. 1a, b, R-1NEA OIHPs have an LHB of 0.181 Å and an SHB of 0.146 Å, while the lengths of LHBs and SHBs in R-2NEA are 0.347 Å and 0.158 Å, respectively. It is well known that the distance from the N atom of the NEA cation to the plane of the axial halogen atom of inorganic framework[33] is a determining factor for estimating the strength of hydrogen-bonding interactions between a spacer cation and the inorganic framework (Supplementary Fig. 2). This means that the mean penetration depth of $NH3^+$ approaching the inorganic framework can be calculated by averaging the length of two different hydrogen bonding (i.e., LHB and SHB) to investigate the effect of the structural isomer cations on hydrogen-bonding interactions in chiral RP OIHPs. Interestingly, despite their structural similarities (they both have the same chiral space group of $P2_1$), R-2NEA OIHP has a significantly larger mean $NH_3^+$ penetration depth (0.253 Å) than R-1NEA OIHP (having a mean penetration depth of 0.164 Å).

The different mean penetration depths of $NH_3^+$ between the isomers were caused by the different locations of the ethylamine groups in the benzene ring (different molecular structures can be recognized in the inset of Fig. 1c), resulting in unequal packing arrangements of the chiral organic molecules (see Fig. 1a, b). As the molecules and the inorganic framework interact via NH···Br hydrogen bonding, the small conformational changes induced by the isomer cations can invoke huge structural modifications in chiral RP OIHPs (different organic molecular packing arrangements). Specifically, the packing structure of the naphthyl moiety can vary significantly depending on the structural isomer, which is due to the considerable steric effect of the double benzene rings. The R-1NEA cations are arranged in slightly recumbent shapes along the in-plane direction of the perovskite lattice (Fig. 1a), while the naphthyl moieties in R-2NEA are arranged vertically along the out-of-plane direction (Fig. 1b). Consequently, the synergetic effect of the

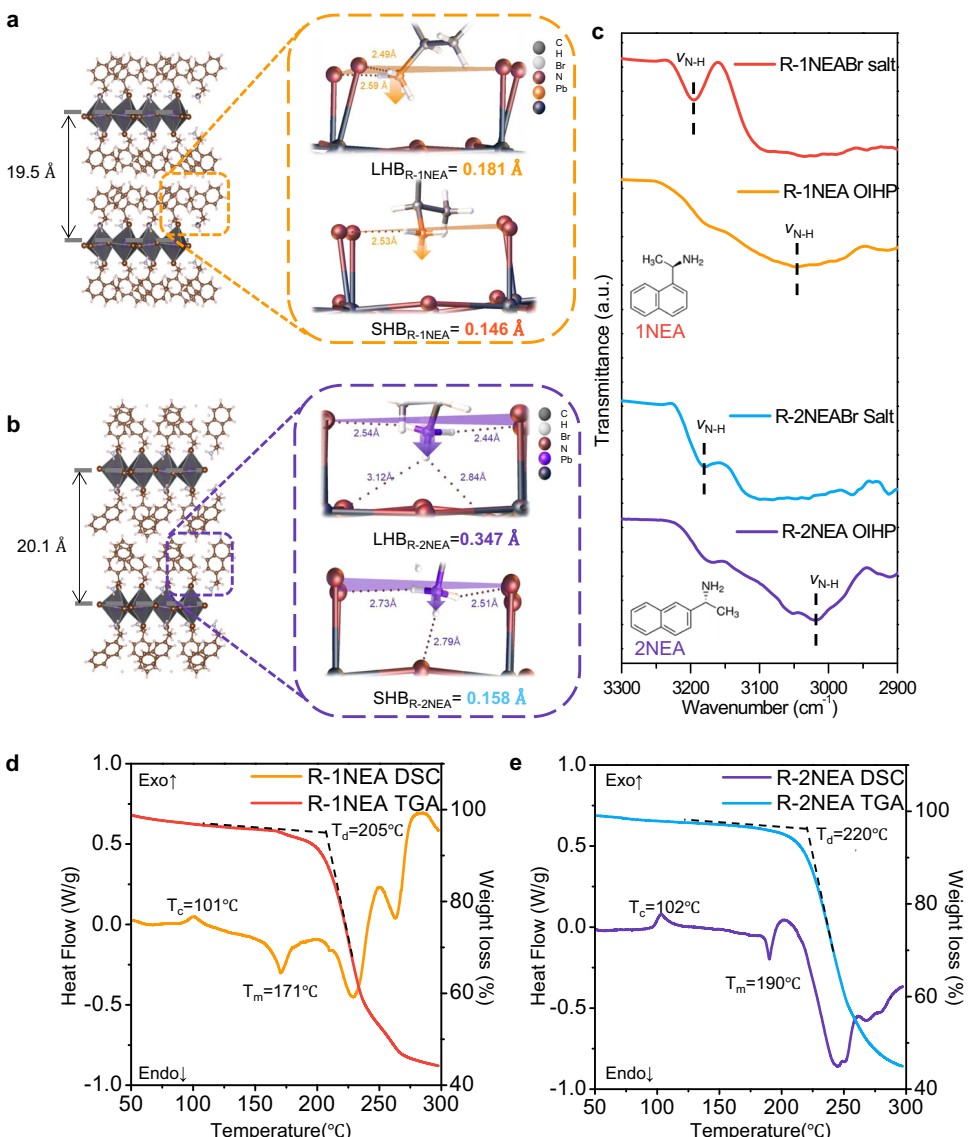

**Fig. 1 | Crystal structure of chiral isomer RP OIHPs and the hydrogen bonding interactions between the chiral organic spacer and the inorganic framework.** Single-crystal structure and asymmetric $NH_3$ penetration depth for **a** (R-1NEA)$_2$PbBr$_4$ and **b** (R-2NEA)$_2$PbBr$_4$. The orange and purple arrows indicate the different penetration depths of terminal amine $NH_3^+$. Gray, white, brown, and dark navy spheres denote C, H, Br, and Pb atoms, respectively. For clarity, the planes of axial Br atom and N atoms are marked in orange and purple for 1NEA and 2NEA, respectively. **c** FT-IR spectra for chiral NEA organic salt and synthesized NEA OIHP powder by transmission mode. Inset: different molecular structures of isomer cations (1NEA and 2NEA). TG-DSC analysis of non-annealed OIHP powder for **d** (R-1NEA)$_2$PbBr$_4$ and for **e** (R-2NEA)$_2$PbBr$_4$. Source data are provided as a Source Data file.

different $NH_3^+$ penetration depth and naphthyl ring arrangements allows *R*-2NEA OIHP to have an elongated interlayer distance of 20.1 Å compared with 19.5 Å for *R*-1NEA OIHP In addition, different $NH_3^+$ penetration depth makes the difference in length between LHBs and SHBs more enlarged in chiral RP OIHPs with the *R*-2NEA isomer (0.189 Å) compared with chiral RP OIHPs with *R*-1NEA (0.035 Å), suggesting that the strength and asymmetric nature of hydrogen-bonding can be enhanced in *R*-2NEA OIHP. To confirm the influence of the structural isomer on the hydrogen-bonding interaction in chiral RP OIHPs, we conduct a Fourier transform infrared (FT-IR) analysis. Figure 1c shows that both chiral RP OIHPs with structural isomers show pronounced N-H symmetry stretch band peaks at wavenumbers of 3047 cm$^{-1}$ and 3053 cm$^{-1}$ for *R*-1NEA and *R*-2NEA OIHPs, respectively, implying that the hydrogen ions in $NH_3^+$ interact fully with inorganic layers via N-H···Br hydrogen bonding[4,34]. The FT-IR result indicated that both chiral NEA

molecules can be successfully intercalated into the lattice of RP OIHPs by forming hydrogen bonding interactions regardless of different functional group positions.

We also performed the thermogravimetric–differential scanning calorimeter (TG-DSC) analysis to compare the strength of hydrogen bonding interactions in chiral RP OIHPs with different isomer chiral cations[27,28]. The structural differences in the isomer cations might give rise to different thermal behavior[35] due to the different $NH_3^+$ penetration depths and coherent hydrogen-bonding interactions. In particular, the melting temperature ($T_m$) of low-dimensional perovskite can be a determinant parameter for estimating the strength of the hydrogen-bonding between bulky cations and inorganic frameworks[27]. As shown in Fig. 1d, e, R-1NEA OIHP has a lower melting temperature ($T_m$) (171 °C) than *R*-2NEA OIHP (190°C), suggesting that the strength of hydrogen bonding interactions was enhanced in chiral RP OIHPs with *R*-2NEA molecule than *R*-1NEA based chiral RP OIHPs. Such an

**Table 1 | Summary of melting temperatures and number of hydrogen bonds**

|  | R-1NEA | R-2NEA |
|---|---|---|
| Melting temperature (°C) | 171 | 190 |
| Average penetration depth (Å) | 0.164 (0.181/0.146) | 0.253 (0.158/0.347) |
| Number of hydrogen bonds | 1 or 2 | 3 |

enhancement of hydrogen-bonding interactions in *R*-2NEA can be explained by the increased number of hydrogen bonds between the chiral spacer and the inorganic framework (Fig. 1a, b, Supplementary Fig. 3, and Table 1). Because of the low steric hindrance effect of 2NEA cations, the functional group in 2NEA can deeply penetrate the inorganic framework, thereby increasing the number of hydrogen bonds between the inorganic framework and the $NH_3^+$ functional group in the 2NEA cations. Consequently, we can infer that the small divergence of molecular structure in the isomer (i.e., the different positions of functional groups) can lead to significant differences in crystallographic parameters and thermal behavior by modulating the hydrogen bonding interactions between the chiral spacer and the inorganic frameworks.

To examine the effect of the chiral isomer cations on chiroptical properties, the RP OIHPs with different NEA isomers in the form of thin films were deposited via the spin-coating process on fluorine-dope tin oxide (FTO) substrates and followed by annealing at 120 °C. Before analyzing the CD measurement, it was necessary to confirm the crystalline structure of the RP OIHPs thin film. We compared the XRD spectra of the thin films with the SC-XRD spectra obtained from the simulated diffraction patterns using VESTA software (Fig. 2a). The main diffraction peaks of the *R*-1NEA OIHP thin-film corresponded to the simulated XRD peak position with 4.54° diffraction peak periodicity, indicating that the fabricated thin film of *R*-1NEA possesses the same crystalline structure as the single crystal. As shown in Fig. 2a, the thin film of RP OIHPs with different structural isomers shows sharp diffraction peaks with no impurity peaks, implying that the (NEA)$_2$PbBr$_4$ OIHPs preferentially grow parallel to the (002 *l*) planes. The XRD spectra from the 1NEA OIHPs show a diffraction peak at 4.72° corresponding to the (002) plane, while the thin film of 2NEA OIHPs shows the diffraction peak of the (002) plane at 4.58°. The thin-film XRD results suggest that the *R*-2NEA OIHPs have a larger interlayer distance than *R*-1NEA OIHP, while the *R*-2NEA thin film shows (002 *l*) diffraction peaks with 4.4° periodicity, which is consistent with the crystallographic study based on the SC-XRD results.

As shown in Fig. 2b, the CD spectra of chiral RP OIHP thin film with different isomer exhibited a distinctive Cotton effect near the first excitonic band edge (~390 nm) regardless of the type of chiral isomer cation, implying that the chirality of NEA molecules was successfully transferred to the inorganic framework. The *R*-1NEA RP OIHP exhibits positive mono-signate peaks centered at 393 nm, while the *R*-2NEA OIHP shows an entirely different CD shape (i.e., bi-signate CD peak centered at 382 nm). The mirror image of the Cotton effect with a similar intensity can be observed in chiral RP OIHPs with an *S*-configuration, while the CD signal is unobserved in racemic counterpart owing to its high-symmetric crystal structure (Supplementary Fig. 4). To quantitatively determine the chiroptical capability of chiral RP OIHPs, the anisotropy factor ($g_{CD}$) was calculated from the CD spectra using equation 1 and presented in Fig. 2c, which is independent of the film thickness, representing the intrinsic chiroptical response[26,36]:

$$g_{CD} = \frac{CD}{(32980 \times \text{absorbance})} \quad (1)$$

where CD and absorbance represent the ellipticity of the CD signal and the intensity from the UV-visible absorption spectroscopy,

respectively. The absolute value of $g_{CD,max}$ was enhanced from $g_{cd,max,R\text{-}1NEA} = 2.01 \times 10^{-3}$ for *R*-1NEA OIHP to $g_{cd,max,2NEA} = -2.78 \times 10^{-3}$ for *R*-2NEA OIHP corresponding to a 38% increase (Fig. 2c), implying that the efficiency of chirality transfer from the chiral spacer to the inorganic framework also depends on the molecular structure of the isomers.

Although the chiral isomer cation-dependent chiroptical phenomena can be clearly recognized in the CD spectra, we cannot conclude that the distinct chiroptical behavior originates from the different molecular structures of isomers (or different hydrogen bonding interactions induced by chiral isomer cations) because of the possible interference resulting from the macroscopic morphological features of thin films. The use of different chiral isomer cations can facilitate different thermal behavior as well as a different crystallization process, leading to the formation of different morphologies in the chiral RP OIHP thin film. Therefore, we initially examined the morphology of the chiral RP OIHP thin film. As shown in Supplementary Fig. 5, the pure *R*-1NEA OIHP thin film exhibits a rough, leaf-like morphology, while the *R*-2NEA has a somewhat smoother morphology than *R*-1NEA. It is well known that the CD signal in thin films with a rough surface or microstructure can be overestimated owing to the experimental measurement limitations of transmitted CD measurements[1] (Supplementary Fig. 6). Instead, to scrutinize the true effect of the structural isomer on spin-related properties of chiral RP OIHPs, we need to exclude the film morphology derived from possible interference. To eliminate the morphological effect on the CD spectra, we added a small amount of methylammonium (MA) in the form of 10 mol% of methylammonium bromide (MABr) into the precursor of the chiral RP OIHPs to diminish the roughness of the thin films, as previously reported[4]. It is worth mentioning that the precursor of MABr was adopted as an additive instead of methylammonium iodide (MAI) in order to prevent halogen exchange in the inorganic layer, which may induce unwanted electronic structure variation in chiral RP OIHPs[29]. Both chiral RP OIHPs (1NEA and 2NEA) added with MA exhibited smooth surfaces with no noticeable differences in morphology (Supplementary Fig. 7). As shown in Fig. 2d, the XRD spectra demonstrated that there is no additional impurity phase or secondary phase (MAPbBr$_3$ or PbBr$_2$ etc.) even in the presence of MABr additive. The UV-visible spectra and steady-state photoluminescence spectra show no shifts in the first excitonic transition (Supplementary Figs. 8 and 9). In addition, the calculated bandgaps of chiral RP OIHPs are 3.57 eV for R-1NEA and 3.66 eV for R-2NEA obtained from the Tauc plot (Supplementary Fig. 10), which are exactly the same values for the samples with MABr additive (controlled) and without MABr (pure). The identical bandgap for the samples with and without additive likely originates from the high volatility of MABr, which can easily evaporate during the annealing process[37–39]. Therefore, MABr additive only affects the thin film morphology by controlling the crystallization kinetics without causing unwanted structural interference, compositional change (A-site alloy) or change of electronic structure.

Consequently, we can examine the true effect of the structural isomer on the chiroptical properties by using chiral RP OIHP thin-film in which the morphology contributions are excluded (hereafter referred to as controlled *R*-1NEA or controlled *R*-2NEA). Compared with CD spectra for thin films without the additive, the CD spectra of thin films with the MABr additive show relatively smaller degrees of ellipticity (mdeg), suggesting that the chiroptical activity of chiral RP thin films was overestimated in the absence of the MABr additive. As shown in Fig. 2e, the controlled *R*-1NEA thin film reveals a bi-signate Cotton effect at 393 nm, while the controlled *R*-2NEA thin film exhibits bi-signate Cotton effect at 384 nm. The calculated $g_{CD}$ values of morphology-controlled films are $-6.84 \times 10^{-4}$ for controlled *R*-1NEA and $-9.61 \times 10^{-4}$ for controlled *R*-2NEA, respectively (Fig. 2f). Although the absolute magnitude of $g_{CD,max}$ is reduced by morphological flattening, the sign conversion phenomena and the different magnitudes

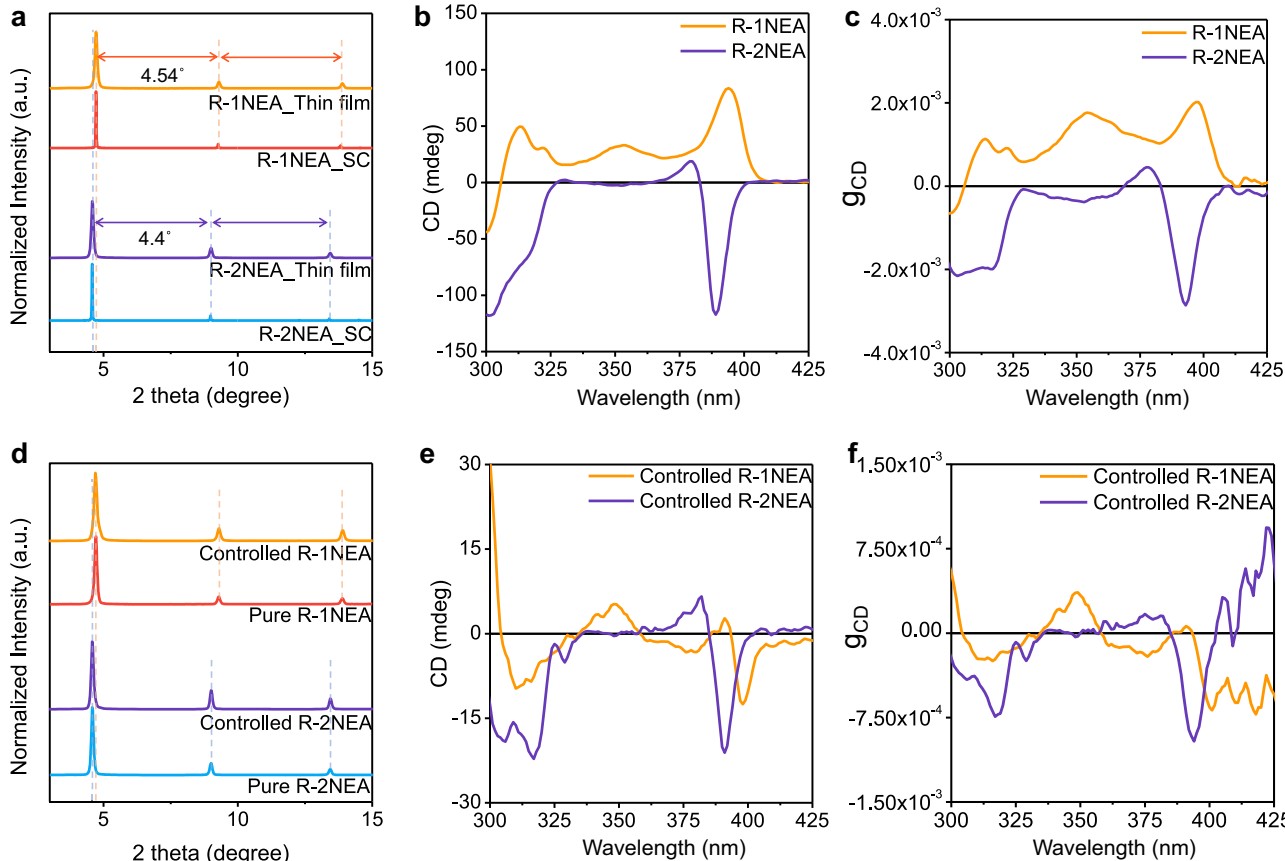

**Fig. 2 | Structural studies and chiroptical activity analysis for NEA isomer OIHP thin films. a** Thin-film XRD pattern of chiral 2D OIHPs and simulated XRD pattern, **b** CD spectra of NEA isomer OIHP thin film, and **c** calculated $g_{CD}$ from the CD spectra. **d** XRD pattern of morphology-controlled and non-controlled (denoted as pure) NEA isomer OIHPs. **e** CD spectra of controlled NEA isomer OIHP thin film, and **f** calculated $g_{CD}$ from the CD spectra. Source data are provided as a Source Data file.

of the chiroptic response depending upon the chiral isomer cation can be clearly observed in the CD spectra of morphology-controlled chiral RP OIHP thin film. It is well known that solid-state thin films with large surface roughness can exhibit unexpected CD signal with a strong dependence on the light propagation direction (incident light direction during the CD measurement with solid-state thin film)[40–42]. The observed optoelectronic behavior stems from the optical interference of thin film's linear birefringence (LB) and linear dichroism (LD) (hereafter LDLB effect), rather than excitonic effects. Many previous studies have reported that huge LDLB effect can contaminate the true chiroptical response in thin film samples with macroscopic roughness. Therefore, we also need to exclude the influence of LDLB contribution to demonstrate the true effect of structural isomer cation on chiroptical activity of chiral RP OIHPs. Since the LDLB effect contribution inverts upon sample flipping (i.e., flipping the sample by 180° with respect to the light propagation axis), the $CD_{true}$ term can be separately obtained by taking semi-sum of the two CD spectra with different measurement directions, (i.e., front and back). As shown in $CD_{true}$ spectra (Supplementary Fig. 11), the thin films of controlled R-2NEA clearly exhibits sign-inversion behavior near the first extinction band edge (around 390 nm) when compared to controlled R-1NEA. Based on the obtained $CD_{true}$ spectra, the effect of the optical anisotropy due to the macroscopic nature can be completely excluded. Therefore, we can conclude that the observed different chiroptical responses of chiral RP OIHPs can be attributable to the effect of the different structural isomer chiral cations rather than to the differences in the morphology of thin films or LDLB effect.

In a previous study, Ben-Moshe et al. demonstrated that the total CD spectra result from the sum of the multiple excitonic transition

peaks in the optical spectrum[43], suggesting a protocol that could determine the magnitude of excitonic state splitting ($\Delta E$) by deconvoluting the CD spectra using Gaussian fitting (Fig. 3a, b). As the magnitude of excitonic state splitting is closely related to the efficiency of chirality transfer[26], $\Delta E$ provides a useful clue for elucidating how the incorporated structural isomer chiral cations regulate the chiroptical activity of overall chiral RP OIHPs. As shown in Fig. 3a, b, we derived the excitonic state splitting values from the deconvoluted CD spectra (see Supplementary Note 1 and Supplementary Fig. 12 for experimental details for peak deconvolution process from the CD spectra). Interestingly, the controlled chiral RP OIHP with different isomers showed different excitonic splitting values (55.8 meV for R-1NEA and 83.6 meV for R-2NEA, respectively), implying that efficient chirality transfer can be facilitated by adopting R-2NEA rather than R-1NEA as a chiral spacer in chiral RP OIHPs.

Although the electronic transition confirmed by CD measurement, where the transition dipoles between the electronic states are included (i.e., electronic transition dipole and magnetic transition dipole), can provide profound information of both of ground state and excited state, the asymmetric absorption behavior and associated CD at first extinction band edge in chiral RP OIHPs can be rationalized by the procedure from the selected subset of thermally relaxed vibrational state of the ground state to variety of vibrational states in excited state. Thus, in principle, CD spectroscopy contains useful information regarding the electronic structure of the excited state in chiral RP OIHPs. As the circularly polarized photoluminescence (CPPL) is based on opposite optical transition phenomena of CD (i.e., emission of light for CPPL and absorption of light for CD), these complementary phenomena can be exploited to establish the profound information

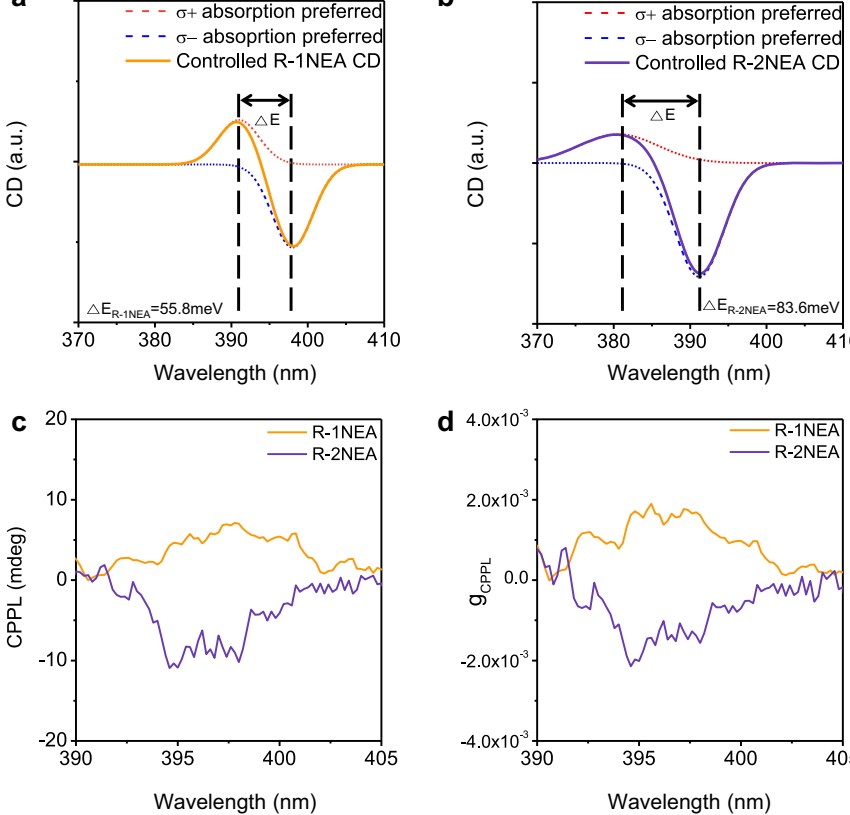

**Fig. 3 | Deconvolution of CD spectra for excitonic state splitting calculation and CPPL spectra.** Deconvolution results were obtained from the CD spectra of controlled NEA isomer OIHPs using Gaussian fitting; **a** *R*-1NEA and **b** *R*-2NEA. **c** CPPL spectra of R-1NEA and R-2NEA OIHPs thin film measured under an excitation source of 345 nm. **d** $g_{CPPL}$ spectra of R-NEA structural isomer OIHPs. Source data are provided as a Source Data file.

about the electronic structure of chiral RP OIHPs. Interestingly, the CPPL spectra for the *R*-1NEA and *R*-2NEA exhibited CPL emission behavior with completely opposite handedness (Fig. 3c) regardless of the fact that both of 1- and 2NEA spacer have the same handedness. The asymmetry factors ($g_{CPPL}$) calculated from the CPPL spectra are $1.89 \times 10^{-3}$ for *R*-1NEA and $-2.14 \times 10^{-3}$ for *R*-2NEA (Fig. 3d and Supplementary Fig. 13). This sign conversion phenomena are similar to that observed in the CD spectra at the first extinction band edge (Fig. 2e). To further investigate the photophysical properties of chiral RP OIHPs, we also conducted a series of experiments. As shown in Supplementary Fig. 8, the first excitonic state is clearly observed 392 nm for 1NEA perovskite and 387 nm for 2NEA perovskite, respectively. The blue shift of first excitonic transition state in 2NEA perovskite is attributed to larger interplanar distance between the inorganic slab (~20.1 Å) compared to 19.5 Å for *R*-1NEA OIHP (Fig. 1). The PL emission spectra of chiral RP OIHPs with different structural isomer also show the difference, with Stoke shifts of ~11 nm. Interestingly, the excitonic peak of the chiral RP OIHPs with 2NEA is sharper and more intense, while chiral RP OIHPs with 1NEA exhibit a broader and weaker excitonic peak. This result implies that the large interplanar distance in chiral RP OIHPs with 2NEA gives rise to the increased dielectric confinement that further facilitates exciton recombination process in the interior lattice. To confirm the luminescence mechanism in chiral RP OIHPs, time-resolved photoluminescence (TRPL) spectroscopy is also conducted (Supplementary Fig. 14). A bi-exponential fitting was used to extract the lifetimes and relevant parameters were presented in Table S4. Both of chiral RP OIHPs have slow lifetime components associated with recombination in the lattice interior in the range of a few nanoseconds (2.068 ns for 1NEA sample and 1.985 ns for 2NEA sample)[7,44]. This reveals that free exciton relaxes very fast due to the large exciton binding energy in 2D chiral RP OIHPs.

To realize the CPPL in chiral materials, a photon emitted from an excitonic transition state should be polarized. For the CPPL in chiral materials, the slight perturbation of energy state should be preceded by the chirality transfer (Supplementary Fig. 15). Although a perturbation of energy state associated with chirality transfer can be acquired through a variety of different mechanisms including overall chiral shape; chiral crystal lattice; chiral surface; chiral defect, erhaps the best pathway to obtain CPPL with strong polarization asymmetry is the chiral exciton that can be generated in the chiral crystal lattice. Based on the results of excitation and emission spectroscopy and crystallographic analysis, we can conclude that the observed CPPL in chiral RP OIHPs originate from the chiral exciton generated in the chiral crystal lattice. In line with the results of CD spectra, the CPPL result also implies that the modulating the hydrogen-bonding interaction between the chiral molecules and inorganic frameworks can facilitate the chirality transfer but also modulate the electronic structure of chiral RP OIHPs, which is consistent with our previous theoretical expectation[26].

Therefore, based on our crystallographic studies and chiroptical spectroscopy, we propose the following plausible stepwise mechanism (Fig. 4) to regulate the chiroptical response of the chiral RP OIHPs by the structural isomers:
(i)   the slight divergence of molecular structure in the isomers can induce the enhanced asymmetric hydrogen-bonding interaction between the chiral cation and $(PbBr_6)^{4-}$ inorganic layer;
(ii)  the lattice distortion of inorganic frameworks is promoted by the enhanced asymmetric hydrogen-bonding interactions;

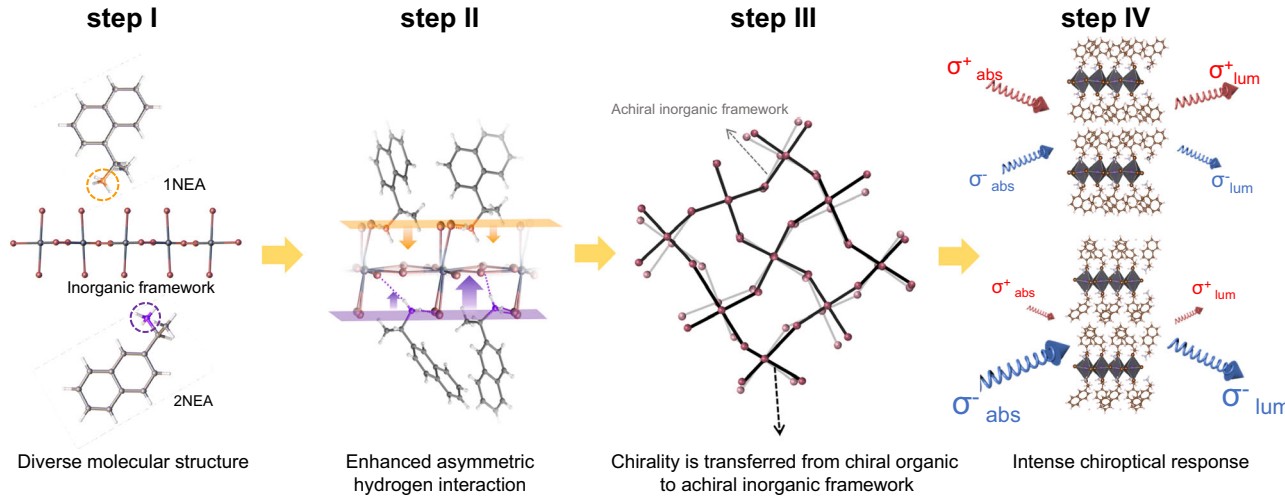

**step I** | **step II** | **step III** | **step IV**

Diverse molecular structure | Enhanced asymmetric hydrogen interaction | Chirality is transferred from chiral organic to achiral inorganic framework | Intense chiroptical response

**Fig. 4 | Schematic illustration of the stepwise chirality transfer mechanism of the structural isomer OIHPs.** The chirality of the organic spacers is efficiently transferred to the inorganic framework by distorting the inorganic layers. The $\sigma_{abs}$ and $\sigma_{lum}$ in step IV represent the absorption and luminescence of the RCP (+) and LCP (−), respectively.

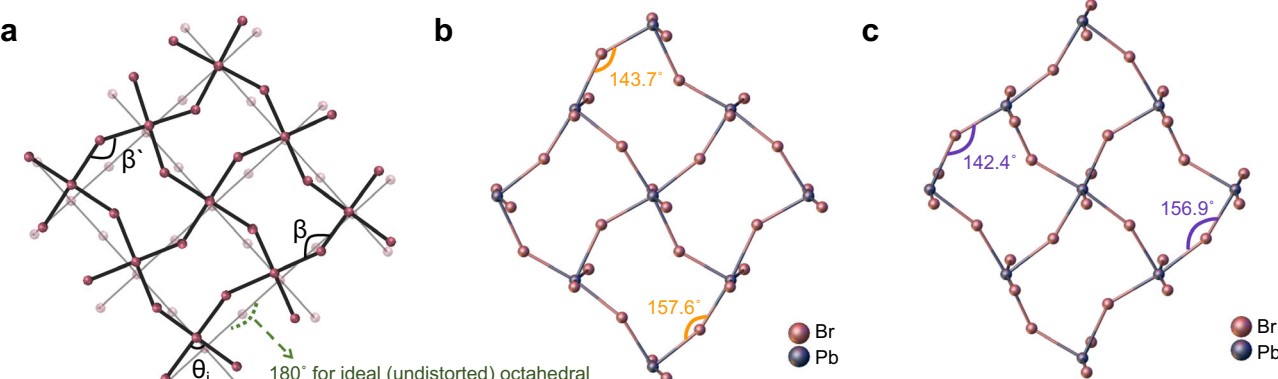

**Fig. 5 | Structural origins of the different degrees of chiroptical response.**
**a** Scheme of distorted inorganic octahedra from ideal inorganic octahedra caused by bulky spacers and angle measurements of β and β'. Note that the $\theta_i$ indicates cis

Br-Pb-Br angles for $\sigma^2$ calculation. Inorganic layer distortion induced by NEA spacers for **b** R-1NEA, and **c** R-2NEA. The inorganic layers are viewed from the [001] direction and the C, H, and N atoms are omitted for clarity.

(iii) the chirality is transferred from the chiral organic spacer to the inorganic framework;

(iv) an intense chiroptical response occurs at the first extinction band edge associated with the excitonic transition in the distorted inorganic framework.

To verify the validity of the proposed mechanism (with respect to the origin of the enhanced chiroptical activity by structural isomer), we calculated the structural parameters based on crystallographic data from SC-XRD. At first, the degree of inorganic octahedral distortion was estimated by comparing the deviation of the Pb-Br-Pb angle away from 180° (undistorted ideal octahedra in Fig. 5a)[20,33]. As shown in Fig. 5b, c, the R-2NEA OIHP has a more significant inorganic octahedral distortion of 37.6° and 23.1° than the R-1NEA OIHP (deviation of 36.4° and 22.4°). To quantify the degree of octahedral distortion, the bond angle variance ($\sigma^2$) was calculated using Eq. (2).

$$\sum^2 = \left(\frac{1}{11}\right)\sum_{i=1}^{12}(\theta_i - 90)^2 \qquad (2)$$

The $\sigma^2$ illustrates that the R-2NEA OIHP has a more severe octahedral distortion than R-1NEA OIHP as shown in Table 2. Furthermore, we also carefully determined the specific structural parameters of the

asymmetric bond angle disparity (Δβ)[24,25], which is known to correlate strongly with the spin-related properties.

Although both R-1NEA and R-2NEA OIHPs exhibit inversion symmetry-broken features (i.e., Pb-Br-Pb tilting angle disparity) because of the asymmetric hydrogen-bonding interaction (as confirmed by the crystallographic study in Fig. 1a, b), the R-1NEA OIHP has a smaller equatorial Pb-Br-Pb angle disparity of 13.9° (i.e., Pb-Br-Pb angles are 157.6° and 143.7°), whereas R-2NEA OIHP has a 14.5° angle disparity (i.e., Pb-Br-Pb angles of 156.9° and 142.4°). It is worth noting that the S-NEA OIHPs with the P2₁ chiral space group exhibit the same tendency (i.e., angle disparity for S-2NEA to be greater than angle disparity for S-1NEA) as similarly observed in R-NEA OIHP (Supplementary Fig. 16 and Table S2). The racemic-NEA OIHPs with P2₁/c global centrosymmetric space groups possess inorganic layers with no Pb-Br-Pb angle disparity, implying that structural chirality transfer cannot occur in the absence of asymmetric hydrogen bonding interactions (Supplementary Fig. 17 and Table S3), which is consistent with our previous report[26]. In other words, the enhanced asymmetric nature of hydrogen-bonding interactions induced by the structural isomer (i.e., 2NEA > 1NEA) can invoke lattice distortion in the inorganic framework, which facilitates the chirality transfer phenomena from the chiral spacer to the inorganic framework. The crystallographic study closely matches the

**Table 2 | Structural parameters for the chiral NEA OIHPs single crystal and the first excitonic peak wavelength**

| | Bond angle variance ($\sigma^2$, deg$^2$) | Equatorial Br-Pb-Br bond angle (β, β') | Asymmetric bond angle disparity (Δβ = β−β') | First excitonic peak wavelength |
|---|---|---|---|---|
| R-1NEA | 40.47 | 157.6°/143° | 13.9° | 392 nm |
| R-2NEA | 48.02 | 156.9°/142.4° | 14.5° | 387 nm |

experimental results, demonstrating the validity of our proposed mechanism.

It is worth mentioning that the chiral organic molecules have their own chirality and also exhibit CD response. As shown in Supplementary Fig. 18, the chiral organic molecules show mirror image of CD signal at ~290 nm depending on their handedness, which is associated with their exciton transition state. However, this transition state is far from the excitonic transition state of chiral RP OIHPs (~380 nm), so that the coupling between the dipole moments of the chiral organic molecules (1NEA or 2NEA) and the transition dipoles of lead-bromide framework is unlikely to occur. Therefore, it can be concluded that the chiroptical response of our chiral RP OIHPs should be interpreted as a result of induced lattice distortion by the chirality transfer phenomena through the asymmetric hydrogen bonding assisted symmetry breaking rather than electronic coupling between two building blocks. To support our conclusion, the degree of lattice distortion in inorganic framework, which is closely related with the chirality transfer efficiency, was investigated by examining the UV-visible absorption spectra of chiral RP OIHPs with different chiral isomer cations. When the lattice of inorganic frameworks is significantly distorted, the overlap of the Pb and Br orbital states decreases, leading to a bandgap-widening in the OIHPs[33]. As shown in Supplementary Fig. 19, the different excitonic peak positions associated with the first excitonic transition state were clearly recognizable (392 nm for R-1NEA and 387 nm for R-2NEA, respectively). The hypsochromic shift of the excitonic peak observed in R-2NEA RP OIHPs confirms that the degree of lattice distortion can be intensified with R-2NEA, facilitating chirality transfer from the chiral spacer molecules to achiral inorganic frameworks. The UV-visible absorption spectra result also clearly affirms that the larger chiroptical activity (both CD and CPPL) in R-2NEA RP OIHPs is originated from the enhanced chirality transfer phenomena (as well as more asymmetric hydrogen-bonding interaction) induced by structural isomer with different functional group location.

As a proof-of-concept, planar-type circularly polarized light-photodetectors (CPL-PDs) based on chiral RP OIHPs was demonstrated. Supplementary Fig. 20 describes the structure of CPL-PDs as well as experimental procedures to investigate its capability to discriminate between LCP and RCP illumination. The photocurrent vs time curve of CPL-PDs with different isomers under LCP and RCP illumination by using laser at 365 nm, 385 nm, and 400 nm were presented in Supplementary Fig. 21. Both of CPL-PDs exhibited reliable operational stability and distinguishability upon repeated illumination measurement. In addition, the photocurrent vs voltage curve of CPL-PDs with different isomer under LCP and RCP illumination by using the same laser applying external voltage range from −4 V to 4 V were also provided (Supplementary Fig. 22). It is worth noting that in CPL-PDs based on R-2NEA always exhibited higher photocurrent response to LCP than RCP (Supplementary Fig. 23), whereas CPL-PDs based R-1NEA showed better response to RCP than LCP over entire voltage range, which is consistent with the observed result of CD spectra at the first extinction band edge. Although the current level of our proof-of-concept CPL-PDs is limited to a few of pA level, we clearly demonstrated that the sign conversion phenomenon in CD spectra can be also recognized as a different preferential photocurrent response of CPL-PDs.

Finally, we carried out a thermal stability test to compare the influence of the chiral isomer cation on the structural integrity of chiral RP OIHPs. Both chiral RP OIHPs with chiral NEA isomer cations showed excellent stability under ambient atmosphere after six weeks, exhibiting no phase degradation or decomposition by XRD spectra (Supplementary Fig. 24). The unexpected environmental stability of chiral RP OIHPs stems from the bromide component, which can effectively prevent oxidation within the perovskite lattice[19]. Therefore, to accurately investigate the effect of the isomer cations on the thermal and moisture stability of the bromide chiral RP OIHPs, the stability test should be performed under harsh conditions of 75 °C and 75% relative humidity (RH). The environmental stability of chiral RP OIHPs was then assessed by tracing the XRD, CD, and UV-visible absorption spectra for 7 days.

As shown in Fig. 6a, the layered structure of chiral RP OIHPs with 1NEA is destroyed after 3 days of storage. The diffraction peak at 4.72°, corresponding to the (002) plane, splits into 3.8° and 5° peaks, while the (002l) peaks with regular periodicity completely disappear. The CD and absorption spectra reveal that the Cotton effect (Fig. 6b) and first extinction band edge corresponding to the excitonic transition state almost disappear after 3 days (Supplementary Fig. 25a) due to the decomposition of perovskite crystal, which is consistent with the XRD spectra. In contrast, despite the phase segregation (from 2D phase to mixed-phase 1D/2D) and slight peak shift of (002) from 4.58° to 4.76°, the layered-2D structure of chiral RP OIHPs with 2NEA maintain even after 7 days (Fig. 6c). The optical absorption spectra show that the extinction peak corresponding to the first excitonic transition state maintains (Supplementary Fig. 25b). Moreover, the Cotton effect associated with the first excitonic transition state is clearly observable even after 7 days of storage (Fig. 6d). The shape changes of the Cotton effect may arise from the mixed structure of 1D and 2D phases[4].

The structural robustness of chiral RP OIHPs based on R-2NEA can be explained by the different hydrogen-bonding nature of the structural isomer chiral cations. As shown in the TG-DSC spectra in Fig. 2d, e, R-1NEA RP OIHP shows relatively lower $T_m$ and degradation temperature ($T_d$, weight loss onset temperature) than its R-2NEA counterpart. Since the lead-bromide framework is connected by rigid primary bonding (i.e., ionic and covalent bonds)[26], the structural degradation and weight loss can be attributed to the breaking of the hydrogen-bonding interaction between the NEA molecules and the [PbBr$_4$]$^{6-}$ octahedra[27,28]. Thus, we can conclude that the superior structural robustness of R-2NEA RP OIHP was caused by the stronger hydrogen bonding between the chiral spacer and inorganic frameworks.

To fully understand the effect of chiral isomer cations on structural stability, the formation energy of chiral RP OIHPs with different structural isomer is evaluated (Fig. 6e). In the initial state, the mixture of NEABr and PbBr$_2$ had a nearly identical total energy value regardless of the isomer cation structure (~0.01 eV). However, in the final state, the chiral RP OIHP with R-2NEA had lower formation energy (−4.38 eV) than the chiral RP OIHP with the R-1NEA isomer (−4.15 eV). The smaller formation energy mainly originates from the lower formation enthalpy due to the stronger hydrogen-bonding interaction and conformational arrangement of bulky naphthyl moiety along the out-of-plane direction in the chiral R-2NEA isomer-incorporated RP OIHPs, as mentioned above[45,46]. Consequently, the crystal structure of R-2NEA OIHP is energetically favorable over R-1NEA OIHPs, demonstrating that the stronger hydrogen bonds in R-2NEA OIHPs are much more stable than R-1NEA OIHPs in harsh environmental conditions.

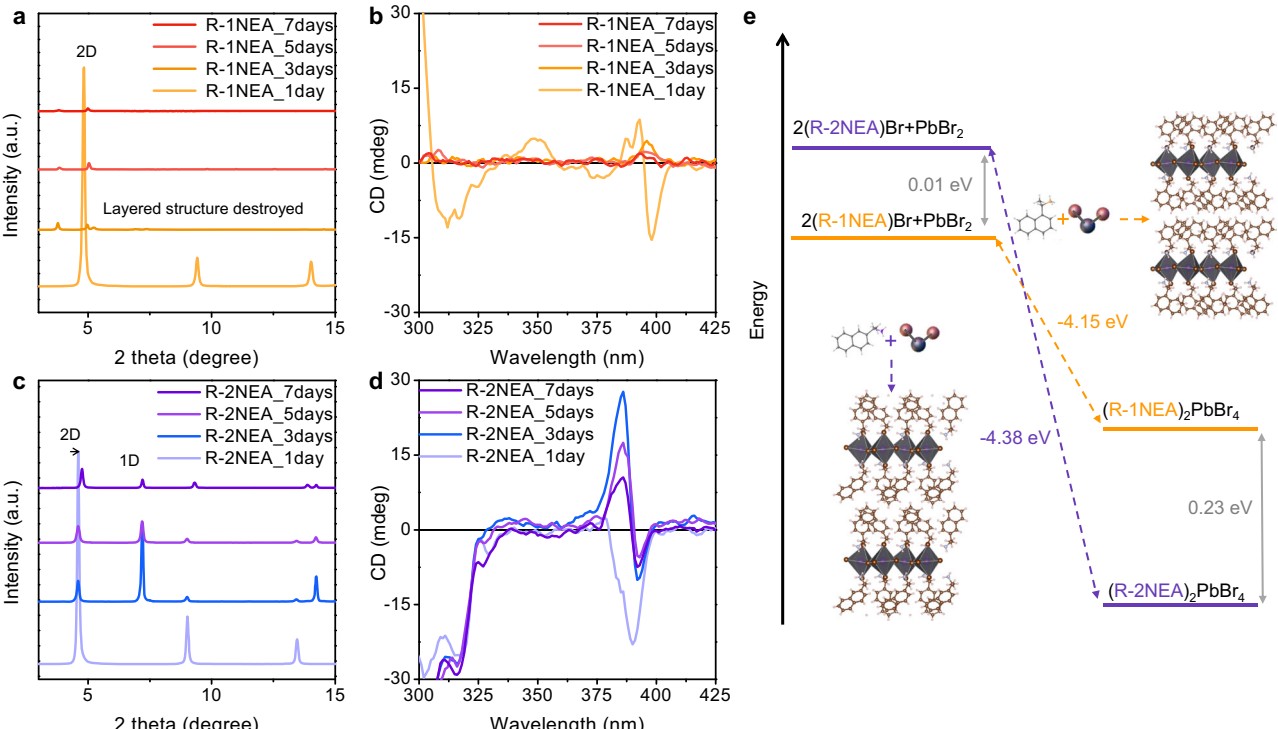

**Fig. 6 | Environmental stability test of the controlled NEA isomer OIHPs.** XRD pattern for 7 days under adverse conditions (75% RH and 75 °C) for **a** *R*-1NEA and **c** *R*-2NEA OIHP thin film. The CD spectra were traced for 7 days **b** *R*-1NEA and **d** *R*-2NEA OIHP thin film under the same stability test conditions. **e** DFT calculation of the relative formation energy difference for the (*R*-NEA)$_2$PbBr$_4$ structural isomer OIHPs. Source data are provided as a Source Data file.

## Discussion

In summary, we manipulated the spin-related properties, structural features, and environmental stability of chiral RP OIHPs by adopting different chiral structural isomer cations. Depending upon the different locations of functional groups (NH$_3^+$), we were able to vary the degree of hydrogen-bonding interactions between the chiral organic cations and the inorganic frameworks. Through detailed experimental verification, we demonstrate the true effect of asymmetric hydrogen-bonding on the spin-related properties of chiral RP OIHPs, which is also related to the modulation of chiroptical activity in chiral RP OIHPs. Based on our experimental results and crystallographic studies, we proposed a stepwise chirality transfer mechanism via asymmetric hydrogen-bonding interactions between chiral spacer cations and inorganic frameworks. The stronger (and more asymmetric) hydrogen-bonding interaction in 2NEA chiral RP OIHP results in a greater level of inversion asymmetric distortion in the inorganic layers, promoting the chirality transfer phenomena from chiral molecules to inorganic frameworks. Consequently, 2NEA chiral RP OIHP exhibited more enhanced spin-polarized photon absorption behavior (~40% increase in anisotropy factor) than 1NEA chiral RP OIHP. Furthermore, we also demonstrated that the sign conversion phenomenon in CD spectra can be converted in a different preferential photocurrent response of CPL-PDs. Moreover, 2NEA chiral RP OIHPs also demonstrated exceptional environmental stability under harsh conditions (75 °C and 75% RH) for 7 days. Our results suggest that small conformational changes in isomers (such as different locations of functional groups) can lead to greatly differing spin-related and structural properties of chiral RP OIHPs. Our proposed strategy based on structural isomer chiral cations clarifies the relationship between hydrogen-bonding interactions and chiroptical phenomena in chiral RP OIHPs. We expect that important applications and next-generation energy conversion devices with excellent performance may be developed based on our preliminary observations and proposed material design rule.

## Methods

### Materials
(S)-(-)-1-(1-naphthyl)ethylamine (>98%), (R)-(+)-1-(1-naphthyl)ethylamine (>98%), (S)-(−)-1-(2-naphthyl)ethylamine (>98%), (R)-(+)-1-(2-naphthyl)ethylamine (>98%) were purchased from TCI chemicals. HBr acid (48 wt% in H$_2$O, >99.99%), PbBr$_2$ (99.999% trace metals basis) DMF (anhydrous, 99.8%), and dichloromethane (anhydrous, >99.8%) were purchased from Sigma-Aldrich and were used as received without further purification.

### Chiral organic ammonium salt protonation process
In total, 9.6 mmol of organic amine (1NEA for 1.544 ml, 2NEA for 1643.9 mg) was dissolved in 4 ml of ethanol followed by stirring. After stirring for 5 min, HBr (10.4 mmol, 1.176 mL) was added, and the clear light brownish solutions were stirred for 12 h at room temperature. The stirred solution was then fully evaporated at 80 °C in a vacuum oven for 2 days and the off-white solid precipitates were washed with diethyl ether several times and dried in a vacuum at 80 °C for 2 days.

### Single-crystal synthesis
(R/S-2NEA)$_2$PbBr$_4$ single crystals were synthesized using the AVC method. Next, the synthesized NEABr and PbBr$_2$ crystals were dissolved in DMF solution at a molar ratio of 2:1 and 50 wt% concentration. The precursor was stirred for 2 h on a hot plate at 35 °C. The brown-colored precursor solutions were then filtered through a 0.2 μm syringe filter to the 20 mL vial. The small vials were transferred to a large 70 mL vial, and 12 mL of DCM was injected into the gap between the 20 mL vial and 70 mL vial. The large vial was fully capped with parafilm and stored in a constant temperature chamber at 23 °C for 6 days (Supplementary Fig. 1). The synthesized transparent block-shaped crystals were filtrated, washed with diethyl ether several times, and dried in a vacuum oven at 35 °C for 2 days. However, the racemic-2NEA OIHPs synthesized by the AVC method exhibited poor quality for SC-XRD measurement. Therefore, the racemic- 2NEA OIHPs were

synthesized by the inverse-temperature cooling (ITC) method, as previously reported[18] (Supplementary Fig. 26). Stoichiometric amounts of $PbBr_2$ (45 mg, 0.12 mmol) and racemic-2NEA (42 mg, 0.24 mmol) were dissolved in a mixture of HBr (0.5 mL) and methanol (1.2 mL). The mixture was heated at 120 °C for 30 min before being slowly cooled to room temperature in a temperature-controlled oven at a rate of 2 °C/min. The resulting colorless, transparent needle-like single crystals were then vacuum filtrated, washed with diethyl ether several times, and dried in a vacuum at 35 °C for 2 days. The crystallographic and structural refinement data of $(R-/S-/Rac-2NEA)_2PbBr_4$ are shown in Table S1–S3.

## Perovskite thin film and powder preparation

Each of the FTO substrates (2 cm × cm) was cleaned using D.I water, acetone, and ethanol in a sonicator for 15 min. Next, they received direct UV-ozone treatment for 15 min before spin-coating. To prepare the precursor, NEABr (0.346 mmol, 86.6713 mg) and $PbBr_2$ (0.173 mmol, 63.3287 mg) were dissolved in DMF (628 μL, 600 mg) to satisfy 20 wt% concentration. After stirring for 4 h at room temperature, the precursor was filtrated and spin-coated onto an FTO substrate at 2000 rpm for 30 s with an amount of 60 μL before the substrates were annealed for 30 min at 120 °C. The powders were prepared by scratching off the coated thick film. The powder samples were prepared by spin-coated using 30 wt% precursors onto the 5 cm × 5 cm soda-lime glass substrate at 1500 rpm for 30 s.

## Characterizaion of chiroptical and optical properties

CD and absorbance spectra were analyzed using a J-815 spectrometer (JASCO Corporation). The baseline was measured in air with a 5 mL/min $N_2$ flow and the scan rate was 200 nm/min with a data pitch of 1 nm. The thin film sample was 2 cm × 1.2 cm, and the coated surface was measured to face the light source. The CPL spectra were analyzed using CPL-300 spectrometer (JASCO Corporation). The scan rate was 100 nm/min with a data pitch of 0.2 nm with 345 nm excitation source. All the spectra were measured under a condition with a maximum DC voltage of -0.5 V (Supplementary Fig. 13). The $g_{CPPL}$ value was directly calculated from the instrument program. Steady-state PL spectra were collected with excitation beam wavelength of 350 nm and TRPL spectra were collected with excitation beam of 371 nm (FluoroMax Plus, Horiba, Kyoto, Japan).

## Single-crystal X-ray diffraction

The single-crystal XRD structure analysis was performed at 298 K on a Bruker SMART APEX-II instrument (Mo Kα radiation, $\lambda = 0.71073$ Å operating at 50 kV and 30 mA). The structural parameters were obtained by SHELXT and refined by SHELXL. The crystallographic structure was then visualized using VESTA and OLEX2 software.

## X-ray diffraction

XRD analysis of the thin film was conducted using Smart Lab (Rigaku Miniflex 600) with a Cu Kα radiation source ($\lambda = 1.54$ Å) and a scan rate of 3°/min. The obtained peaks were calibrated with the FTO substrate peaks (26.6°, 33.8°, and 37.86°) to accurately compare the peak position.

## Device fabrication of CPL-PDs and characterization

The soda-lime galss substrate was cleaned as above mentioned. After UV-ozone treatment, the 20 wt% precursors was spin-coated onto the substrate at 2000 rpm for 30 s before annealing for 30 min at 120 °C. After the annealing, 70 nm thick Au top electrode was deposited onto the thin films with thermal evaporation. The CPL photocurrent vs time curves and photocurrent vs voltage curves of PDs were measured at a bias voltage of 4 V using an Agilent 4155 C in a probestation.

## Stability test

The harsh environmental stability test was conducted using a humidity- and temperature-controlled chamber (TH-PE-025, JEIO Tech, Korea). The unencapsulated, prepared thin film OIHPs were maintained in an environment of 75 °C and 75% RH.

## DFT calculation

By adopting the experimentally determined crystal structures as initial input structures, we optimized the crystal structures of OHIPs on the basis of the Kohn–Sham density-functional theory (DFT)[47] and obtained DFT total energies from the optimized ground state structures. The underlying DFT calculations were performed using the Vienna Ab Initio Simulation Package (VASP)[48,49] where Projector augmented-wave (PAW)[50,51] pseudopotentials were employed to treat core states. The valence states of H, C, N, Pb, and Br are treated explicitly by 1(1s1), 4(2s22p2), 5(2s22p3), 14(5d106s26p2), 7(4s24p5) electrons, respectively. The Perdew–Burke–Ernzerhof exchange-correlation functional (PBE)[52] with the Grimme *D3*[53] scheme for van der Waals corrections was adopted where the plane-wave kinetic energy cutoff and Γ-centered k-mesh were set to of 700 eV and 6 × 6 × 2. The convergence criteria for total energy and atomic forces were set to 10−6 eV and 10−3 eVÅ−1, respectively. NEABr molecules and Pnma $PbBr_2$ were regarded as reference states when calculating the formation energies.

## FT-IR and TG-DSC measurement

The FT-IR spectra were measured using a Vertex 70 (Bruker) transmission mode with a small amount of perovskite powder (-1.0 mg) contained in a KBr pellet. The TG-DSC measurement was simultaneously conducted using SDT Q600 (TA instrument) with a powder sample (-5.0 mg) in an alumina pan. The heat ramping rate was 5 °C/min, from 30 °C to 300 °C, after the samples were isothermally heated at 30 °C for 1 min.

## Surface morphology characterization

The morphologies of the OIHPs surface were examined using a field-emission SEM (JSM-7001F, JEOL). The AFM topography was analyzed using a SPA 400 (Seiko Instruments, Inc.) in an area of 40 μm × 40 μm with a scan rate of 1 Hz with an Rh-coated cantilever (spring constant, -1.8 N/m).

## Data availability

All data generated or analyzed during this study along with its Supplementary Information are included in this published article. Source data are provided as a Source Data file. The crystal structure of (R/S/Rac-1NEA)$_2$PbBr$_4$ is redrawn based on the Cambridge Crystallographic Data Centre (CCDC); R-1NEA: 2015620 (https://doi.org/10.5517/ccdc. csd.cc25nf09), S-1NEA: 2015618 (https://doi.org/10.5517/ccdc.csd. cc25ndy6), Rac-1NEA: 2015614 (https://doi.org/10.5517/ccdc.csd. cc25ndt2). The cif data of (R/S/Rac-2NEA)$_2$PbBr$_4$ single crystal structure analyzed in this work are accessible in CCDC; R-2NEA:2178604 (https://doi.org/10.5517/ccdc.csd.cc2c40k3), S-2NEA: 2178605 (https:// doi.org/10.5517/ccdc.csd.cc2c40l4), Rac-2NEA: 2178606 (https://doi. org/10.5517/ccdc.csd.cc2c40m5). Source data are provided with this paper.

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

## Acknowledgements
This work was supported by a National Research Foundation (NRF) of Korea grant (2018M3D1A1058793 and 2021R1A3B1068920) funded by the Ministry of Science and ICT. This research was also supported by the Yonsei Signature Research Cluster Program of 2021 (2021-22-0002).

## Author contributions
J.S. and S.M. conceived the idea. The fabrication process of the experimental samples was developed by J.S., C.U.L., H.L., and G.J. under the supervision of J.T. and S.M. J.S. and S.M. designed the study. J.S., J.L., S.M., and W.J. performed optical experiments and analyzed data under the supervision of S.M. and J.M. Y.-K.J. and A.W. performed the density-functional theory calculations. S.M. and J.M. wrote the manuscript with contributions from all other co-authors.

## Competing interests
The authors declare no competing interests.
