## [Peer review file · Nature Communications]

REVIEWER COMMENTS

Reviewer #1 (Remarks to the Author):

This work synthesized lead halide based chiral perovskite materials with two structural isomers of naphthyl ethylamines. Different stereochemistry of the organic amines contributed to different H-bonding interactions to halides, i.e. numbers and strengths, that further resulted in lattice distortions on the perovskite crystal structures in chiral manner. It was interesting to see the simple change in molecular structure could affect overall chiroptical performances of CD and CPPLs. The logics and interpretations on the experimental results are sound and could contribute to the understanding the molecular designs of ligands for chiral perovskite materials. Therefore, I recommend this article after major revisions. Details are described in following.

- 1) Introduction. Please include why the chiral perovskite materials are important and what are their advantages for the suggested applications.
- 2) Figure 1. It is difficult to observe locations and numbers of H-bondings for each cases. It is hard to link the descriptions on the numbers and lengths of H-bondings in the manuscript to the enlarged drawings of Figure 1a, b. Please consider to add other views for clear views on the bonding sites. Also difficult to notice the meaning of the arrows in the Figure.
- 3) R or S-NEAs have their own chirality. In addition to surface distortions of the lead-halide crystal lattices, is there any possibility that coupling between the dipole moments of the organic ligands and the transition dipoles of the semiconducting perovskite crystals contributes to the chiroptical properties?
- 4) About the discussions on the surface roughness of the perovskite thin films. As authors indicated, surface roughness could raise apparent CD signals in transmission mode due to contributions of scattering effects from the surface, especially for the CD measurements on the solid materials. However, probably effects of the surface roughnesses are too exaggerated. Diffused reflectance CD (DRCD) may provide more direct evidences for this.
- 5) Page 10. It is difficult to agree on the Authors descriptions, "the sign conversion phenomena and the different magnitudes of the chiroptic response depending upon the chiral isomer cation can be clearly observed in the CD spectra of morphology-controlled chiral RP OIHP thin film." CD of controlled R-1NEA (Figure 2e) showed very weak CD signals and totally different to CD of the pure R-1NEA of Figure 2b. It seems that maximum CD peak is inversed with opposite sign and shifted to higher wavelength. Such weak signals and inversions could be observed since the CD are very sensitive and subtle to the molecular structural changes. Please consider to reconsider this part with more elaboration. Probably due to weaker H-bonding interactions and more bulky stereochemistry of 1NEA than 2NEA, addition of 10 mol% may be too much to interfere molecular arrangements and surface distortions than expected. Investigations with samples with additive contents less than 10 mol% are recommended for clear observations on the change of CD spectra.

6) Supplementary Fig.11 does not have absorbance spectra, different to the caption title. Some locations have typo errors in super or subscripts like NH_3^+ .

Reviewer #2 (Remarks to the Author):

In this manuscript, using structural isomers with different functional group positions, the author can infer the influence of the hydrogen bond interaction of two isomers on the degree of chiral transfer in the inorganic framework. This is the first time to prove the effect of asymmetric hydrogen bond interaction on chiral transfer. The method of controlling the asymmetry of hydrogen bond interaction through the small structural difference of isomer cations can provide a reasonable design paradigm for the unprecedented spin related properties of chiral perovskite. The manuscript is well written, but there are still some problems to be solved.

1. Although the arrangement of isomers eliminates the influence of components, the position of different functional groups will also bring about the stereo structure effect. How to eliminate this effect?

2. In addition, the research progress of chiral perovskite mixed with A-site alloy introduced by the author is still insufficient, such as *J. Phys. Chem. Lett.* 2021, 12, 12129.

From the CD spectrum, the perovskite composed of the same isomer is different, and the steady-state spectrum should also be different. The reviewer believes that the changes in the three-dimensional structure caused by the same isomer cannot be ignored. Does the author have a more reasonable explanation?

Reviewer #3 (Remarks to the Author):

This manuscript reports studies of the chirality transfer by structural isomer-derived hydrogen bonding interaction in 2D chiral perovskites. The presented work and results are sufficiently innovative and noteworthy. These studies could also be quite important for the research field of chiral materials. However, the manuscript can not be published in present form and requires a major revision. The following issues must be addressed.

The materials are anisotropic and have 2D morphology. Therefore, some linear dichroism (LD) studies are necessary as the LD effects can contribute to the chiroptical activity of these materials and this must be taken into account.

There is no proper luminescence studies of the proposed materials except CPPL spectroscopy. There are no ordinary excitation and emission spectra of materials, no luminescent life-times measurements and no any explanation of luminescence mechanisms and their origin in this materials. Without all the presented CPPL results are not clear and irrelevant.

There is no any proper conclusions in the manuscript and no any outlook on the potential applications of the research. Without that the manuscript does not look complete.

Response Letter

<Reviewer 1>

This work synthesized lead halide based chiral perovskite materials with two structural isomers of naphthyl ethylamines. Different stereochemistry of the organic amines contributed to different H-bonding interactions to halides, i.e. numbers and strengths, that further resulted in lattice distortions on the perovskite crystal structures in chiral manner. It was interesting to see the simple change in molecular structure could affect overall chiroptical performances of CD and CPPLs. The logics and interpretations on the experimental results are sound and could contribute to the understanding the molecular designs of ligands for chiral perovskite materials. Therefore, I recommend this article after major revisions. Details are described in following.

Remark:

We would like to thank the reviewer for evaluating our work. Our response to the reviewer's comments can be found below.

Comment 1:

Introduction. Please include why the chiral perovskite materials are important and what are their advantages for the suggested applications.

Author's Response:

We entirely agree with the reviewer that it is necessary to explain why the chiral perovskite materials can be considered as a new type of semiconductor for next-generation optoelectronic [R1,R2,R3,R4] and spintronic applications.[R5,R6,R7,R8] Therefore, we have added a paragraph in the Introduction to comment on the excellent properties and advantages of chiral perovskite for the promising applications as reviewer suggested. We thank the reviewer for careful comment.

References cited in this response:

- R1 Galkowski, K. *et al.* Determination of the exciton binding energy and effective masses for methylammonium and formamidinium lead tri-halide perovskite semiconductors. *Energy Environ. Sci.* 2016, **9**, 962.

- R2 Dong, Q. F. *et al.* Electron-hole diffusion lengths > 175 μ m in solution-grown CH₃NH₃PbI₃ single crystals. *Science* 2015, **347**, 967
- R3 Stranks, S. D *et al.* Electron-hole diffusion lengths exceeding 1 micrometer in an organometal trihalide perovskite absorber. *Science* 2013, **342**, 341
- R4 Xing, G. C *et al.* Long-range balanced electron- and hole-transport lengths in organic-inorganic CH₃NH₃PbI₃. *Science* 2013, **342**, 344.
- R5 Zhang, C. *et al.* Magnetic field effects in hybrid perovskite devices. *Nat. Phys.* 2015, **11**, 428.
- R6 Isarov, M. *et al.* Rashba effect in a single colloidal CsPbBr₃ perovskite nanocrystal detected by magneto-optical measurement. *Nano Lett.* 2017, **17**, 5020.
- R7 Mosconi, E., Etienne, T., & Angelis, F. D. Rashba band splitting in organohalide lead perovskites: Bulk and surface effects. *J. Phys. Chem. Lett.* 2017, **8**, 2247
- R8 Odenthal, P. *et al.* Spin-polarized exciton quantum beating in hybrid organic-inorganic perovskites. *Nat. Phys.* 2017, **13**, 894.

Revision made (colored in blue):

(Introduction)

... Incorporating organic molecules with inherent chirality in the A`-site leads to the formation of RP OIHPs having a chiral crystal structure. Although the structural difference of RP OIHPs induced by the enantiomers is small, it can give rise to a huge difference in their spin-related properties. For example, our group initially reported that chiral RP OIHPs that utilized two different enantiomers (*R*- and *S*-methyl benzylamine (*R*- and *S*-MBA)) could exhibit circular dichroism (CD), with an opposite sign near the first excitonic band edge depending on the handedness of the MBA cation¹.

Since the chiroptical response of chiral RP OIHPs was first reported in 2017 by our group,¹ chiral RP OIHPs have been extensively investigated in chiral photonics research society. ~~various spin and polarization based applications such as circularly polarized light (CPL) detection²⁻⁶ and emission⁷⁻⁹, spin filtering¹⁰⁻¹⁴, non-linear optic¹⁵, and ferroelectrics^{16,17} because of their unprecedented spin related properties.~~ Chiral RP OIHPs are based on spontaneously self-assembled multi quantum well (MQW) structure consisting of two alternating building blocks of metal-halide inorganic slab (wells) and bulky chiral organic

spacer (barriers). Therefore, chiral RP OIHPs can exhibit both excellent optoelectronic properties derived from the inorganic slab and unique chiroptical properties induced by the bulky chiral organic spacer. Owing to their stable and broad wavelength tunability over visible light region as well as small effective mass,^[R1] long spin lifetime exceeding 1 ns, diffusion lengths ~ 85 nm, and high electron/hole mobility,^[R2,R3,R4] chiral RP OIHPs can be an ideal candidate for optoelectronic devices based on circularly polarized light (CPL) detection and emission. Furthermore, strong spin-orbit coupling of the electronic state,^[R5] large Rashba-Dresselhaus splitting,^[R6,R7,R8] and spin-dependent optical selection rules in chiral RP OIHPs suggest that chiral RP OIHPs can be also utilized into spin polarization-based devices such as spin filter and ferroelectrics.

However, despite of their excellent spin-related properties, the origins of the spin-polarization-based phenomena (*e.g.*, spin-polarized photon absorption and spin-polarized photoluminescence) is still elusive. ...

(References; relevant references were added)

- R1 Galkowski, K. *et al.* Determination of the exciton binding energy and effective masses for methylammonium and formamidinium lead tri-halide perovskite semiconductors. *Energy Environ Sci.* 2016, **9**, 962.
- R2 Dong, Q. F. *et al.* Electron-hole diffusion lengths > 175 μm in solution-grown $\text{CH}_3\text{NH}_3\text{PbI}_3$ single crystals. *Science* 2015, **347**, 967
- R3 Stranks, S. D. *et al.* Electron-hole diffusion lengths exceeding 1 micrometer in an organometal trihalide perovskite absorber. *Science* 2013, **342**, 341
- R4 Xing, G. C. *et al.* Long-range balanced electron- and hole-transport lengths in organic-inorganic $\text{CH}_3\text{NH}_3\text{PbI}_3$. *Science* 2013, **342**, 344.
- R5 Zhang, C. *et al.* Magnetic field effects in hybrid perovskite devices. *Nat. Phys.* 2015, **11**, 428.
- R6 Isarov, M. *et al.* Rashba effect in a single colloidal CsPbBr_3 perovskite nanocrystal detected by magneto-optical measurement. *Nano Lett.* 2017, **17**, 5020.
- R7 Mosconi, E., Etienne, T., & Angelis, F. D. Rashba band splitting in organohalide lead perovskites: Bulk and surface effects. *J. Phys. Chem. Lett.* 2017, **8**, 2247
- R8 Odenthal, P. *et al.* Spin-polarized exciton quantum beating in hybrid organic-inorganic perovskites. *Nat. Phys.* 2017, **13**, 894.

Comment 2:

Figure 1. It is difficult to observe locations and numbers of H-bondings for each cases. It is hard to link the descriptions on the numbers and lengths of H-bondings in the manuscript to the enlarged drawings of Figure 1a, b. Please consider to add other views for clear views on the bonding sites. Also difficult to notice the meaning of the arrows in the Figure.

Author's Response:

We appreciate the reviewer for important comment regarding the clarity of schematic illustration. The different number of hydrogen bonding between achiral inorganic frameworks and chiral spacer depending on the molecular structure of organic molecules is our core finding and objective in this paper. Therefore, as reviewer suggested, we have revised the schematic description of hydrogen bonding interaction for more clear recognition. In addition, we have replaced Figure 1 with an enlarged and clear image of crystalline structure (Fig. R1) to precisely show the different location and different number of hydrogen bonding between two building blocks. In this schematic side-view illustration along the *c* direction, the orange and purple arrows indicate the different penetration depth of terminal amine NH_3^+ , which varies with the different functional group location (*i.e.*, 1-NEA versus 2-NEA). Furthermore, the magnified crystal structure images of chiral RP OIHPs in the top-view along the *ab*-plane direction have been also added (Fig. R2). We appreciate the reviewer for his/her helpful comment.

Fig. R1. Crystal structure of chiral NEA isomer RP OIHPs and the hydrogen bonding interactions between the chiral organic spacer and the inorganic framework. Single crystal structures and asymmetric NH_3 penetration depth for **a**, $(R-1NEA)_2\text{PbBr}_4$ and **b**, $(R-2NEA)_2\text{PbBr}_4$. The orange and purple arrows represent the degree of penetration depth. The orange and purple arrows indicate the different penetration depth of terminal amine NH_3^+

Fig. R2. Crystal structures of NEA isomer OIHPs viewed from [001] direction.

Revision made (colored in blue):

(Fig. 1a and b were changed)

Fig. 1. Crystal structure of chiral NEA isomer RP OIHPs and the hydrogen bonding interactions between the chiral organic spacer and the inorganic framework. Single crystal structures and asymmetric NH_3 penetration depth for **a**, $(R-1NEA)_2PbBr_4$ and **b**, $(R-$

$2\text{NEA})_2\text{PbBr}_4$. The orange and purple arrows represent the degree of penetration depth. The orange and purple arrows indicate the different penetration depth of terminal amine NH_3^+

(Supplementary; Supplementary Fig.3 was added)

Supplementary Fig. 3. Crystal structures of NEA isomer OIHPs viewed from [001] direction.

Comment 3:

R or S-NEAs have their own chirality. In addition to surface distortions of the lead-halide crystal lattices, is there any possibility that coupling between the dipole moments of the organic ligands and the transition dipoles of the semiconducting perovskite crystals contributes to the chiroptical properties?

Author's Response:

We appreciate the reviewer for constructive comments regarding the electronic coupling between the two building blocks. We understood that reviewer asks us to consider the possible chirality transfer mechanism other than the induced lattice distortion (not surface distortion in our case). In our chiral RP OIHPs system, the chiroptical response (CD and CPPL) is not originated from the R or S-NEA cation exciton transition, but rather associated with the excitonic transition state in lead-bromide framework. As shown in Fig. R3, the R and S-NEA cation itself shows CD response due to their own chirality, which is associated with their exciton transition at ~ 290 nm. This exciton transition of chiral molecules is far from the wavelength where the chiral RP OIHPs excitonic transition occurs (~ 380 nm), so that the coupling between the dipole moments of the chiral organic molecules (1-NEA or 2-NEA) and the transition dipoles of lead-bromide framework is unlikely to occur. Furthermore, based on previous reports, the DFT calculation results demonstrated that the conduction band minimum (CBM) and the valence band maximum (VBM) are mainly determined by the metal 6*p* orbitals and the hybridization of the halide 5*p* and metal 6*s* orbital states.[R9,R10] HOMO or LUMO level of NEA molecules cannot form a resonant state with lead-bromide frameworks due to their huge band offset. In this manner, we can conclude that the chiroptical response of our chiral RP OIHPs should be interpreted as a result of induced lattice distortion by the chirality transfer phenomena through the asymmetric hydrogen bonding assisted symmetry breaking rather than electronic coupling between two building blocks.

Fig. R3. CD and absorbance spectra of chiral NEA molecules. a, CD spectra for R/S-1NEA molecules and **b,** R/S-2NEA molecules. **c,** Absorbance spectra for NEA isomer molecules.

References cited in this response:

- R9 Ma, S. *et al.* Elucidating the origin of chiroptical activity in chiral 2D perovskites through nano-confined growth. *Nat. Commun.* **13**, 3259 (2022).
- R10 Lu, H. *et al.* Highly distorted chiral two-dimensional tin iodide perovskites for spin-polarized charge transport. *J. Am. Chem. Soc.* **142**, 13030-13040 (2020).

Revision made (colored in blue):

(in Page 15~16)

... The crystallographic study closely matches the experimental results, demonstrating the validity of our proposed mechanism.

It is worth mentioning that the chiral organic molecules have their own chirality and also exhibit CD response. As shown in Supplementary Fig. 18, the chiral organic molecules show mirror image of CD signal at ~ 290 nm depending on their handedness, which is associated with their exciton transition state. However, this transition state is far from the excitonic transition state of chiral RP OIHPs (~ 380 nm), so that the coupling between the dipole moments of the chiral organic molecules (1-NEA or 2-NEA) and the transition dipoles of lead-bromide framework is unlikely to occur. Therefore, it can be concluded that the chiroptical response of our chiral RP OIHPs should be interpreted as a result of induced lattice distortion by the chirality transfer phenomena through the asymmetric hydrogen bonding assisted symmetry breaking rather than electronic coupling between two building blocks. To support our conclusion, the degree of lattice distortion in inorganic framework, which is closely

related with the chirality transfer efficiency, ~~can be estimated~~ was investigated by examining the UV-visible absorption spectra of chiral RP OIHPs with different chiral isomer cations. ...

(Supplementary; Fig. 18 was added)

Supplementary Fig. 18. CD and absorbance spectra of chiral NEA molecules. a, CD spectra for R/S-1NEA molecules and **b,** R/S-2NEA molecules. **c,** Absorbance spectra for NEA isomer molecules.

Comment 4:

About the discussions on the surface roughness of the perovskite thin films. As authors indicated, surface roughness could raise apparent CD signals in transmission mode due to contributions of scattering effects from the surface, especially for the CD measurements on the solid materials. However, probably effects of the surface roughnesses are too exaggerated. Diffused reflectance CD (DRCD) may provide more direct evidences for this.

Author's Response:

We appreciate the reviewer for comments regarding the possible experimental artifact for the CD measurement with solid state thin film. It is well known that organic thin films with large surface roughness can exhibit unexpected CD signal with a strong dependence on the light propagation direction (incident light direction during the CD measurement with solid state thin film)^[R11,R12,R13]. The observed optoelectronic behavior stems from the optical interference of thin film's linear birefringence (LB) and linear dichroism (LD) (hereafter LDLB effect), rather than excitonic effects. Therefore, when we investigate the chiroptical activities of thin films with large surface roughness, a basic concept of Mueller matrix analysis is needed to recall;

because the observed transmission CD signal (CD_{obs}) is the sum of various contributions, which can be expressed by the equation (1):

$$CD_{obs} \approx CD_{true} + \frac{1}{2}(LD' \cdot LB - LD \cdot LB') \quad (1)$$

where the first term refers to genuine CD (CD_{true}), while the second term accounts for LDLB effect contribution (the signal of which is taken along an arbitrary axis defined in the laboratory frame in which the prime indicates a 45° axis rotation)^[R11]. Many previous studies have reported that significant contribution of LDLB effect can contaminate the true chiroptical response in thin film samples with macroscopic roughness. Therefore, we need to exclude the influence of LDLB contribution to demonstrate the true effect of structural isomer cation on chiroptical activity of chiral RP OIHPs. Since the LDLB effect contribution inverts upon sample flipping (*i.e.*, flipping the sample by 180° with respect to the light propagation axis), the CD_{true} and LDLB contribution term can be separately obtained by taking semi-sum and semi-difference of the two CD spectra with different measurement directions, (*i.e.*, front and back).

$$CD_{true} = 0.5 \times (CD_{obs, front} + CD_{obs, back}) \quad (2)$$

$$LDLB = 0.5 \times (CD_{obs, front} - CD_{obs, back}) \quad (3)$$

To eliminate the undesirable contamination from the LDLB effect, we have additionally conducted the CD measurement with chiral RP OIHPs thin films by varying the incident light direction. Interestingly, as shown in Fig. R4a and b, both of chiral RP OIHPs thin films without surface morphology control (no MABr added) exhibited the huge CD signal regardless of the light propagation direction (*i.e.*, front and back). However, these huge CD signal almost canceled out upon sample flipping, implying the huge LDLB effect in our chiral RP OIHPs thin films. Furthermore, we isolated CD_{true} and LDLB contribution by using equation (2) and (3) to derive the true effect of structural isomer (*i.e.*, sign conversion of CD signal). As shown in CD_{true} and g-factor of CD_{true} spectra (Fig. R5), the thin film of *R*-2NEA clearly exhibits sign-inversion behavior near the first extinction band edge (around 375 nm) when compared to *R*-1NEA. Based on the obtained CD_{true} spectra, the effect of the optical anisotropy due to the macroscopic nature can be completely excluded from our experimental results, interpretation, and conclusion.

Furthermore, we have also conducted diffused transmission CD (DTCD) by using integrating sphere as reviewer suggested. During the transmission CD measurement, the intensity of the LCP and RCP at detector can differ due to scattering effects from the surface

of thin films. As the integrating sphere can collect all the scattered light, the contribution of light scattering effect on CD signal is excluded (Fig. R6). It is worth noting that LDLB contribution also exists during the DTCD measurement due to the intrinsic asymmetric nature of thin films. Therefore, we conducted sample flipping measurement and isolated the LDLB contribution in the DTCD spectra by using equation (2) and (3) to derive the true DTCD signal ($DTCD_{true}$) of thin films. As expected, the $DTCD_{true}$ signal of chiral RP OIHPs thin films (R-1NEA and R-2NEA) showed no noticeable change when the MABr additive was added for surface morphology control (Fig. R7). This result implies that the chiral RP OIHPs thin films did not exhibit circular differential scattering (CDS) behavior at this wavelength region. Therefore, we can conclude that our observation of the CD signal in the transmission mode does not originate from light scattering due to surface roughness, but from the excitonic transition in the chiral RP OIHP.

Fig. R4. The CD spectra measured under sample flipping condition. The dashed sky-blue and red lines represent the measured CD spectra from front and back side of the films, respectively. The solid orange and purple lines indicate the CD_{true} , which is calculated by taking semi-sum.

Fig. R5. a, The CD_{true} spectra and **b,** calculated g -factor from CD_{true} of the chiral R-NEA isomer OIHP thin films without additive.

Fig. R6. Schematic illustration of DTCD measurement condition. The integrating sphere collects all the diffused transmission CPL light.

Fig. R7. The DTCD_{true} spectra of NEA isomer OIHP thin films with different MABr concentrations. The DTCD_{true} was calculated by taking semi-sum of front and back side DTCD spectra.

References cited in this response:

- R11 Salij, A., Goldsmith R. H. & Tempelaar, R. Theory of apparent circular dichroism reveals the origin of inverted and noninverted chiroptical response under sample flipping. *J. Am. Chem. Soc.* **143**, 21519-21531 (2021).
- R12 Albano, G. et al. Outstanding chiroptical features of thin films of chiral oligothiophenes. *ChemNanoMat* **4**, 1059-1070 (2018).
- R13 Albano, G., Lissia M., Pescitelli G., Aronica L. A. & Di Bari, L. Chiroptical response inversion upon sample flipping in thin films of a chiral benzo[1,2-b:4,5-b']-dithiophene-based oligothiophene. *Mater. Chem. Front.* **1**, 2047-2056 (2017).

Revision made (colored in blue):

(in Page 11)

... Although the absolute magnitude of $g_{CD,max}$ is reduced by morphological flattening, the sign conversion phenomena and the different magnitudes of the chiroptic response depending upon the chiral isomer cation can be clearly observed in the CD spectra of morphology-controlled chiral RP OIHP thin film. ~~These results infer that the observed different chiroptical responses~~

~~of chiral RP OIHPs can be attributable to the effect of the different structural isomer chiral cations rather than to the differences in the morphology of thin films.~~

It is well known that solid-state thin films with large surface roughness can exhibit unexpected CD signal with a strong dependence on the light propagation direction (incident light direction during the CD measurement with solid state thin film)⁴²⁻⁴⁴. The observed optoelectronic behavior stems from the optical interference of thin film's linear birefringence (LB) and linear dichroism (LD) (hereafter LDLB effect), rather than excitonic effects. Many previous studies have reported that huge LDLB effect can contaminate the true chiroptical response in thin film samples with macroscopic roughness. Therefore, we also need to exclude the influence of LDLB contribution to demonstrate the true effect of structural isomer cation on chiroptical activity of chiral RP OIHPs. Since the LDLB effect contribution inverts upon sample flipping (*i.e.*, flipping the sample by 180° with respect to the light propagation axis), the CD_{true} term can be separately obtained by taking semi-sum of the two CD spectra with different measurement directions, (*i.e.*, front and back). As shown in CD_{true} spectra (Supplementary Fig. 11), the thin films of controlled *R*-2NEA clearly exhibits sign-inversion behavior near the first extinction band edge (around 390 nm) when compared to controlled *R*-1NEA. Based on the obtained CD_{true} spectra, the effect of the optical anisotropy due to the macroscopic nature can be completely excluded. Therefore, we can conclude that the observed different chiroptical responses of chiral RP OIHPs can be attributable to the effect of the different structural isomer chiral cations rather than to the differences in the morphology of thin films or LDLB effect.

In a previous study, Ben-Moshe *et al.* demonstrated that the total CD spectra result from the sum of the multiple excitonic transition peaks in the optical spectrum³¹, ...

(References; relevant references were added)

- 42 Salij, A., Goldsmith, R. H. & Tempelaar, R. Theory of apparent circular dichroism reveals the origin of inverted and noninverted chiroptical response under sample flipping. *J. Am. Chem. Soc.* **143**, 21519-21531 (2021).
- 43 Albano, G. *et al.* Outstanding chiroptical features of thin films of chiral oligothiophenes. *ChemNanoMat* **4**, 1059-1070 (2018).
- 44 Albano, G., Lissia M., Pescitelli G., Aronica L. A. & Di Bari, L. Chiroptical response inversion upon sample flipping in thin films of a chiral benzo[1,2-b:4,5-b']-dithiophene-based oligothiophene. *Mater. Chem. Front.* **1**, 2047-2056 (2017).

(Supplementary; Supplementary Fig. 11 was added)

Supplementary Fig. 11. The CD spectra measured under sample flipping condition. The dashed sky-blue and red lines represent the measured CD spectra from front and back side of the films, respectively. The solid orange and purple lines indicate the CD_{true} , which is calculated by taking semi-sum.

Comment 5:

Page 10. It is difficult to agree on the Authors descriptions, “the sign conversion phenomena and the different magnitudes of the chiroptic response depending upon the chiral isomer cation can be clearly observed in the CD spectra of morphology-controlled chiral RP OIHP thin film.” CD of controlled R-1NEA (Figure 2e) showed very weak CD signals and totally different to CD of the pure R-1NEA of Figure 2b. It seems that maximum CD peak is inversed with opposite sign and shifted to higher wavelength. Such weak signals and inversions could be observed since the CD are very sensitive and subtle to the molecular structural changes. Please consider to reconsider this part with more elaboration. Probably due to weaker H-bonding interactions and more bulky stereochemistry of 1NEA than 2NEA, addition of 10 mol% may be too much to interfere molecular arrangements and surface distortions than expected. Investigations with samples with additive contents less than 10 mol% are recommended for clear observations on the change of CD spectra.

Author's Response:

We appreciate the reviewer for constructive and critical comments regarding main concern of our investigation. As reviewer commented, the addition of excess additive may cause an unwanted change in the stacking order of chiral organic molecules, leading to unexpected lattice distortion when intercalated into the lattice of chiral RP OIHPs. To examine the effect of excess additive (MABr) on the structural change in chiral RP OIHPs, the crystalline structure of chiral RP OIHPs as a function of MABr additive concentration has been scrutinized by using XRD analysis. As shown in Fig. R8, all the chiral RP OIHPs with R-1NEA have shown similar XRD patterns, regardless of different MABr additive concentrations. In detail, the main diffraction peak at 4.72° corresponding to the (002) plane does not shift with the addition of MABr, while the 4.54° regular periodicity of (002) peaks is completely maintained. Furthermore, there is no impurity phase or secondary phase (MAPbBr_3 or PbBr_2) in the XRD spectra even with the addition of 20 mol% MABr additive. This observation demonstrates that excess MABr does not intercalate into the lattice of chiral RP OIHPs. It is well known that excess MABr additive can be easily removed during the annealing process due to its high volatility^[R14,R15,R16]. Therefore, we can conclude that adding MABr does not cause an unwanted effect on the electronic or crystal structure of chiral RP OIHPs, although it does affect the film morphology by controlling the crystallization kinetics.

As the reviewer pointed out, the CD spectra of controlled R-1NEA (Figure 2e in our submitted manuscript) showed totally different behavior compared with CD spectra of the pure R-1NEA in Figure 2b (in our submitted manuscript). However, in the CD_{true} spectra in Fig. R9a obtained from the sample flipping measurement (following the procedure aforementioned in our response to **comment 1** for *Reviewer 1*), the CD signal of R-1NEA at the first extinction band edge (~ 390 nm) does not change depending on the concentration of MABr additive. Genuine CD signal in the CD_{true} spectra (Fig. R9a and b) is very similar to the CD response of controlled R-1NEA and R-2NEA in Fig. R9c (reproduced from Fig. 2e in our submitted manuscript). This implies that the CD response of R-1NEA in Fig.2b (in our submitted manuscript) was exaggerated by the LDLB and morphological effect, whereas the CD response of controlled R-1NEA in Fig. 2e (in our submitted manuscript) is close to genuine CD signal due to reduced thin film roughness by the MABr additive. These results confirm that volatile MABr is suitable additive for obtaining reliable CD signal by only affecting the morphology of thin films without causing unwanted interference or lattice distortion.

Fig. R8. Normalized XRD spectra for morphology-controlled NEA OIHPs thin films. XRD pattern of OIHPs thin films with various MABr additive concentrations for **a**, R-1NEA and **b**, R-2NEA. Note that peak shift or impurity peaks are not observed even if excess MABr is added.

Fig. R9. CD_{true} spectra of morphology-controlled NEA OIHPs thin films. The true-CD spectra are calculated by taking semi-sum of the two CD spectra with different measurement directions (*i.e.*, front and back).

References cited in this response:

- R14 Wu, H. *et al.* Methylammonium bromide assisted crystallization for enhanced lead-free double perovskite photovoltaic performance. *Adv. Funct. Mater.* 2022, **32**, 2109402.
- R15 Yang, M. *et al.* Effect of non-stoichiometric solution chemistry on improving the performance of wide-bandgap perovskite solar cells. *Mater. Today Energy.* 2018, **7**, 232-238.
- R16 Fei, C. *et al.* Controlled growth of textured perovskite films towards high performance solar cells. *Nano Energy.* 2016, **27**, 17-26.

Revision made (colored in blue):

(in Page 10)

... As shown in Fig. 2d, the XRD spectra demonstrated that there is no additional impurity phase or secondary phase (MAPbBr₃ or PbBr₂ etc.) even in the presence of MABr additive. The UV-visible and steady-state photoluminescence spectra show no shifts in the first excitonic transition (Supplementary Fig.8 and Supplementary Fig.9). In addition, the calculated bandgaps of chiral RP OIHPs are 3.57 eV for R-1NEA and 3.66 eV for R-2NEA obtained from the Tauc plot (Supplementary Fig.10), which are exactly the same values for the samples with MABr additive (controlled) and without MABr (pure). The identical bandgap for the samples with and without additive likely originates from the high volatility of MABr, which can easily evaporate during the annealing process³⁹⁻⁴¹. Therefore, MABr additive only affects the thin film morphology by controlling the crystallization kinetics without causing unwanted structural interference, compositional change (A-site alloy) or change of electronic structure. ~~indicating that adding MABr does not cause an unwanted effect on the electronic or crystal structure of chiral RP OIHPs, although it does affect the film morphology by controlling the crystallization kinetics.~~

Consequently, we can examine the true effect of the structural isomer on the chiroptical properties by using chiral RP OIHP thin-film in which the morphology contributions are excluded (hereafter referred to as controlled R-1NEA or controlled R-2NEA). ...

(References; relevant references were added)

- 39 Wu, H. *et al.* Methylammonium bromide assisted crystallization for enhanced lead-free double perovskite photovoltaic performance. *Adv. Funct. Mater.* 2022, **32**, 2109402.

- 40 Yang, M. *et al.* Effect of non-stoichiometric solution chemistry on improving the performance of wide-bandgap perovskite solar cells. *Mater. Today Energy*. 2018, **7**, 232-238.
- 41 Fei, C. *et al.* Controlled growth of textured perovskite films towards high performance solar cells. *Nano Energy*. 2016, **27**, 17-26.

Comment 6:

Supplementary Fig.11 does not have absorbance spectra, different to the caption title. Some locations have typo errors in super or subscripts like NH_3^+ .

Author's Response:

Thank you for the comment. We have checked thoroughly the entire our manuscript in the submitted manuscript and have corrected the mistakes which are not precise or ambiguous statements.

Revision made (colored in blue):

(Supplementary Information)

Supplementary Fig. 16. Crystal structure ~~and absorbance spectra~~ of the S-NEA structural isomer OIHPs viewed from various directions. **a** and **b**, Crystal structure of S-NEA structural isomer OIHPs viewed from [100] direction; **a**, (S-1NEA)₂PbBr₄ **b**, (S-2NEA)₂PbBr₄. **c** and **d**, Inorganic layer structure viewed from [001] direction; **c**, (S-1NEA)₂PbBr₄ and **d**, (S-2NEA)₂PbBr₄. Brown and dark spheres denote Br and Pb atoms, respectively. C, H, and N atoms are omitted for clarity. **E**, Normalized absorbance spectra for (S-NEA)₂PbBr₄.

Supplementary Fig. 17. Crystal structure and absorbance spectra of the racemic-NEA structural isomer OIHPs viewed from various directions. **a** and **b**, Crystal structure of racemic-NEA structural isomer OIHPs viewed from [100] direction; **a**, (Rac-1NEA)₂PbBr₄ and **b**, (Rac-2NEA)₂PbBr₄. **c** and **d**, Inorganic layer structure viewed from [001] direction; **c**, (Rac-1NEA)₂PbBr₄ and **d**, (Rac-2NEA)₂PbBr₄. Brown and dark spheres denote Br and Pb atoms, respectively. C, H, and N atoms are omitted for clarity.

<Reviewer 2>

In this manuscript, using structural isomers with different functional group positions, the author can infer the influence of the hydrogen bond interaction of two isomers on the degree of chiral transfer in the inorganic framework. This is the first time to prove the effect of asymmetric hydrogen bond interaction on chiral transfer. The method of controlling the asymmetry of hydrogen bond interaction through the small structural difference of isomer cations can provide a reasonable design paradigm for the unprecedented spin related properties of chiral perovskite. The manuscript is well written, but there are still some problems to be solved.

Remark:

We would like to gratefully thank the reviewer for reviewing and evaluating our work. We believe that the reviewer's comments highly improve the quality of our manuscript. Our response to the reviewer's comments can be found below.

Comment 1:

Although the arrangement of isomers eliminates the influence of components, the position of different functional groups will also bring about the stereo structure effect. How to eliminate this effect?

Author's Response:

We appreciate the reviewer's comment. However, it is unnecessary to eliminate the stereo structure effect derived from different functional group locations between 1-NEA and 2-NEA. The main and core objective of this paper is to elucidate **the effect of the stereo structural differences in chiral organic spacer on the overall crystal structure and the associated chiroptical response of chiral RP OIHPs**. As expected, the very tiny structural difference between two isomers (different functional group location) can give rise to the huge different stereo structural effect, resulting in completely different three-dimensional (3D) spatial stacking of chiral organic molecules in the lattice of chiral RP OIHPs. Consequently, the different numbers of hydrogen bonding between achiral inorganic framework and chiral spacer are induced depending on the molecular structure of isomer (please refer to Fig. R1 in our response to **comment 1** for *Reviewer 1*).

Comment 2:

In addition, the research progress of chiral perovskite mixed with A-site alloy introduced by the author is still insufficient, such as *J. Phys. Chem. Lett.* 2021, 12, 12129.

From the CD spectrum, the perovskite composed of the same isomer is different, and the steady-state spectrum should also be different. The reviewer believes that the changes in the three-dimensional structure caused by the same isomer cannot be ignored. Does the author have a more reasonable explanation?

Author's Response:

We appreciate the reviewer's critical comment. We understood that reviewer asks us about the origin of different chiroptical behavior (CD spectra, Fig. 2b and e in our submitted manuscript) in the presence of MABr additive. As the reviewer commented, if this different chiroptical response is truly originated from the different excitonic transition behavior of chiral RP OIHPs induced by the A-site alloy, the steady-state photoluminescence (PL) spectra should be also different. However, as shown in Fig. R10, the steady-state PL emission wavelength had no shift upon the addition of MABr, indicating that the different chiroptical response is not attributed to different excitonic transition behavior or different electronic structure of chiral RP OIHPs. In addition, we have estimated the optical bandgap of chiral RP OIHPs obtained from the Tauc plot. As shown in Fig. R11, the calculated bandgaps of chiral RP OIHPs are 3.57 eV for R-1NEA and 3.66 eV for R-2NEA, which are exactly the same values with MABr additive (controlled) and without additive (pure). Based on these results, as mentioned in our response to **comment 4** for *Reviewer 1*, we can conclude that the different CD spectra between the pure- and controlled-chiral RP OIHPs result from the optical interference of thin film's linear birefringence (LB) and linear dichroism (LD) (LDLB effect) rather than change in the three-dimensional structure (A-site alloy). Furthermore, it is well known that volatile MABr additive used for morphology control in our manuscript can be easily removed during the annealing process. Therefore, it does not evoke any compositional change (A-site alloy) or unwanted impurity phase, which is confirmed by XRD spectra (please see our response to **comment 4** for *Reviewer 1*).

Fig. R10. Steady-state PL spectra for NEA isomer OIHP thin films. The excitonic PL peaks are observed at **a**, 404 nm for R-1NEA and **b**, 398 nm for R-2NEA, respectively.

Fig. R11. Tauc plot of the NEA and controlled NEA isomer OIHP thin films. **a**, Tauc plot for R-1NEA and **b**, R-2NEA OIHPs.

Revision made (colored in blue):

(in Page 10)

... As shown in Fig. 2d, the XRD spectra demonstrated that there is no additional impurity phase or secondary phase (MAPbBr₃ or PbBr₂ etc.) even in the presence of MABr additive. The UV-visible and steady-state photoluminescence spectra show no shifts in the first excitonic transition (Supplementary Fig.8 and Supplementary Fig.9). In addition, the calculated bandgaps of chiral RP OIHPs are 3.57 eV for R-1NEA and 3.66 eV for R-2NEA obtained from the Tauc plot (Supplementary Fig.10), which are exactly the same values for the samples with MABr additive (controlled) and without MABr (pure). The identical bandgap for the samples with and without additive likely originates from the high volatility of MABr, which can easily evaporate during the annealing process³⁹⁻⁴¹. Therefore, MABr additive only affects the thin film morphology by controlling the crystallization kinetics without causing unwanted structural interference, compositional change (A-site alloy) or change of electronic structure. ~~indicating that adding MABr does not cause an unwanted effect on the electronic or crystal structure of chiral RP OIHPs, although it does affect the film morphology by controlling the crystallization kinetics.~~

Consequently, we can examine the true effect of the structural isomer on the chiroptical properties by using chiral RP OIHP thin-film in which the morphology contributions are excluded (hereafter referred to as controlled R-1NEA or controlled R-2NEA).

(Supplementary; Supplementary Fig. 9 and Fig.10 were added)

Supplementary Fig. 9. Steady-state PL spectra for NEA isomer OIHP thin films. The excitonic PL peaks are observed at **a**, 404 nm for R-1NEA and **b**, 398 nm for R-2NEA, respectively.

Supplementary Fig. 10. Tauc plot of the NEA and controlled NEA isomer OIHP thin films. **a**, Tauc plot for R-1NEA and **b**, R-2NEA OIHPs.

(experimental; relevant experimental sections were added)

(in Page 22)

~~CD and CPPL analysis~~ Characterization of Chiroptical and Optical properties

CD and absorbance spectra were analyzed using a J-815 spectrometer (JASCO Corporation). The baseline was measured in air with a 5 mL/min N₂ flow and the scan rate was 200 nm/min with a data pitch of 1 nm. The thin film sample was 2 cm × 1.2 cm, and the coated surface was measured to face the light source. The CPL spectra were analyzed using CPL-300 spectrometer (JASCO Corporation). The scan rate was 100 nm/min with a data pitch of 0.2 nm with 345 nm excitation source. All the spectra were measured under a condition with a maximum DC voltage of ~0.5 V (Supplementary Fig. 13). The g_{CPPL} value was directly calculated from the instrument program. *Steady-state PL spectra were collected with excitation beam wavelength of 350 nm and TRPL spectra were collected with excitation beam of 371 nm (FluoroMax Plus, Horiba, Kyoto, Japan).*

<Reviewer 3>

This manuscript reports studies of the chirality transfer by structural isomer-derived hydrogen bonding interaction in 2D chiral perovskites. The presented work and results are sufficiently innovative and noteworthy. These studies could also be quite important for the research field of chiral materials. However, the manuscript cannot be published in present form and requires a major revision. The following issues must be addressed.

Remark:

We would like to gratefully thank the reviewer for reviewing and evaluating our work. We believe that the reviewer's comments highly improve the quality of our manuscript. Our response to the reviewer's comments can be found below.

Comment 1:

The materials are anisotropic and have 2D morphology. Therefore, some linear dichroism (LD) studies are necessary as the LD effects can contribute to the chiroptical activity of these materials and this must be taken into account.

Author's Response:

We appreciate the reviewer's comment. As the reviewer suggested, the observed chiroptical behavior may be exaggerated or contaminated from the optical interference of thin film's linear birefringence (LB) and linear dichroism (LD) (LDLB effect). To eliminate the possibility of undesirable contamination from the LDLB effect, we have additionally conducted the CD measurement with chiral RP OIHPs thin films by varying the incident light direction (*i.e.*, front and back). Diffused transmission CD (DTCD) spectra are also conducted. Please see our response to comment 4 for *Reviewer 1*.

Comment 2:

There is no proper luminescence studies of the proposed materials except CPPL spectroscopy. There are no ordinary excitation and emission spectra of materials, no luminescent life-times measurements and no any explanation of luminescence mechanisms and their origin in this materials. Without all the presented CPPL results are not clear and irrelevant.

Author's Response:

We appreciate the reviewer's critical comment. As reviewer suggested, we additionally conducted a series of experiments including ordinary UV-visible absorption spectroscopy, steady-state PL emission spectra, and time-resolved photoluminescence (TRPL) to further investigate the photophysical properties of chiral RP OIHPs. As shown in Fig. R11a, the first excitonic state is clearly observed at 392 nm for 1NEA perovskite and 387 nm for 2NEA perovskite, respectively. The blue shift of first excitonic transition state in 2NEA perovskite is attributed to larger interplanar distance between the inorganic slab (~ 20.1 Å) compared to 19.5 Å for *R*-1NEA OIHP. In Fig. R10 (in our response to comment 1 for *Reviewer 1*), the PL emission spectra of chiral RP OIHPs with different structural isomer also show the difference, with Stoke shift of ~ 11 nm. Interestingly, the excitonic peak of the chiral RP OIHPs with 2NEA is sharper and more intense, while chiral RP OIHPs with 1NEA exhibit a broader and weaker excitonic peak. This result implies that the large interplanar distance in chiral RP OIHPs with 2NEA gives rise to the increased dielectric confinement that further facilitates

exciton recombination process in the interior lattice. To confirm the luminescence mechanism in chiral RP OIHPs, TRPL spectroscopy is also conducted (Fig. R12). A bi-exponential fitting was used to extract the lifetimes and relevant parameters as presented in Table R1. Both of chiral RP OIHPs have slow lifetime components associated with recombination in the interior lattice in the range of a few nanoseconds (2.068 ns for 1NEA sample and 1.985 ns for 2NEA sample). This reveals that free exciton relaxes very fast due to the large exciton binding energy in 2D chiral RP OIHPs^[R17,R18].

To realize the circularly polarized emission (circularly polarized photoluminescence (CPPL) in our study), a photon emitted from an excitonic transition state should be polarized. For the CPPL in chiral materials, the slight perturbation of energy state should precede the chirality transfer (Fig. R13). Although a perturbation of energy state associated with chirality transfer can be acquired through a variety of different mechanisms including overall chiral shape; chiral crystal lattice; chiral surface; chiral defect, perhaps the best pathway to obtain CPPL with strong polarization asymmetry is the chiral exciton generated in the chiral crystal lattice. Based on the results of excitation and emission spectroscopy and crystallographic analysis, we can conclude that the observed CPPL in chiral RP OIHPs originate from the chiral exciton generated in the chiral crystal lattice. Due to the different degrees of hydrogen bonding interaction accompanying the different degree of chirality transfer, the strong intensity and polarization asymmetry can be obtained in chiral RP OIHPs with 2-NEA isomer. We gratefully thank the reviewer for helpful comments which significantly improve the quality of our manuscript.

Fig. R12. TRPL spectroscopy of R-NEA and controlled R-NEA OIHP thin films measured with excitation wavelength of 371nm. The traces fitted to a biexponential decay function of $y(t) = A_1 \exp(-t/\tau_1) + A_2 \exp(-t/\tau_2)$ with $\text{adj. } R^2 > 0.99$.

Table R1. Fitting parameters of TRPL decay of NEA isomer incorporated OIHP thin films.

	A1 (%)	τ_1	A2 (%)	τ_2	Adj. R^2
R-1NEA	97.08	0.280	2.92	2.068	0.992
Controlled R-1NEA	97.86	0.289	2.14	2.794	0.99
R-2NEA	65.60	0.519	34.40	1.985	0.996
Controlled R-2NEA	62.66	0.47	37.34	1.857	0.997

Fig. R13. Schematic illustration of energy diagram perturbation by chirality and the CPPL emission mechanism. The conduction band minimum (CBM) and valence band maximum (VBM) are composed of the states with total angular momentum quantum number ($J = \frac{1}{2}$) and magnetic quantum number (m_s) of $\pm\frac{1}{2}$ according to the spin-state of electrons. The σ^+_{abs} and σ^-_{abs} indicate the absorption of the RCP (+) and LCP(-), respectively. The σ^+_{abs} corresponds to the excitonic transition where the magnetic quantum number changes from $-\frac{1}{2}$ to $+\frac{1}{2}$ vice versa for σ^-_{abs}). The excitonic states can be perturbed when the chirality is transferred to the OIHP lattice. In this case, the energy states with up-spin state ($m_s = +\frac{1}{2}$) and down-spin state ($m_s = -\frac{1}{2}$) are no longer identical. Therefore, the photon emitted from the chiral OIHPs can be spin-polarized.

References cited in this response:

- R17 Chae, W.-, Cho, S., Jung, J.-, Kim, J.-, & Lee, J.-. Multiple-route recombination dynamics and improved stability of perovskite quantum dots by plasmonic photonic crystal., *J. Phys. Chem. Lett.* 2022, **13**, 5040-5048.
- R18 Nuzzo, D. *et al.* Circularly polarized photoluminescence from chiral perovskite thin films at room temperature. *ACS Nano* 2020, **14**, 7610-7616

Revision made (colored in blue):

(in Page 12-13)

... As the circularly polarized photoluminescence (CPPL) is based on opposite optical transition phenomena of CD (*i.e.*, emission of light for CPPL and absorption of light for CD), these

complementary phenomena can be exploited to establish the profound information about the electronic structure of chiral RP OIHPs.

Interestingly, As shown in Fig. 3c, the CPPL spectra for the *R*-1NEA and *R*-2NEA exhibited CPL emission behavior with completely opposite handedness (Fig. 3e) regardless of the fact that both of 1- and 2-NEA spacer have the same handedness. The asymmetry factors (g_{CPPL}) calculated from the CPPL spectra are 1.89×10^{-3} for *R*-1NEA and -2.14×10^{-3} for *R*-2NEA (Fig. 3d). This sign conversion phenomena are similar to that observed in the CD spectra at the first extinction band edge (Fig. 2e). To further investigate the photophysical properties of chiral RP OIHPs, we also conducted a series of experiments. As shown in Supplementary Fig. 8, the first excitonic state is clearly observed 392 nm for 1NEA perovskite and 387 nm for 2NEA perovskite, respectively. The blue shift of first excitonic transition state in 2NEA perovskite is attributed to larger interplanar distance between the inorganic slab ($\sim 20.1 \text{ \AA}$) compared to 19.5 \AA for *R*-1NEA OIHP (Fig. 1). The PL emission spectra of chiral RP OIHPs with different structural isomer also show the difference, with Stoke shifts of $\sim 11 \text{ nm}$. Interestingly, the excitonic peak of the chiral RP OIHPs with 2NEA is sharper and more intense, while chiral RP OIHPs with 1NEA exhibit a broader and weaker excitonic peak. This result implies that the large interplanar distance in chiral RP OIHPs with 2NEA gives rise to the increased dielectric confinement that further facilitates exciton recombination process in the interior lattice. To confirm the luminescence mechanism in chiral RP OIHPs, time-resolved photoluminescence (TRPL) spectroscopy is also conducted (Supplementary Fig. 13). A bi-exponential fitting was used to extract the lifetimes and relevant parameters were presented in Table S4. Both of chiral RP OIHPs have slow lifetime components associated with recombination in the lattice interior in the range of a few nanoseconds (2.068 ns for 1NEA sample and 1.985 ns for 2NEA sample). This reveals that free exciton relaxes very fast due to the large exciton binding energy in 2D chiral RP OIHPs^{7,46}.

To realize the CPPL in chiral materials, a photon emitted from an excitonic transition state should be polarized. For the CPPL in chiral materials, the slight perturbation of energy state should be preceded by the chirality transfer (Supplementary Fig. 14). Although a perturbation of energy state associated with chirality transfer can be acquired through a variety of different mechanisms including overall chiral shape; chiral crystal lattice; chiral surface; chiral defect, perhaps the best pathway to obtain CPPL with strong polarization asymmetry is the chiral exciton that can be generated in the chiral crystal lattice. Based on the results of excitation and emission spectroscopy and crystallographic analysis, we can conclude that the

observed CPPL in chiral RP OIHPs originate from the chiral exciton generated in the chiral crystal lattice. In line with the results of CD spectra, the CPPL result also implies that the modulating the hydrogen-bonding interaction between the chiral molecules and inorganic frameworks can facilitate the chirality transfer but also modulate the electronic structure of chiral RP OIHPs, which is consistent with our previous theoretical expectation²⁰.

(Reference; relevant references were added)

- 46 Chae, W-, Cho, S., Jung,J-,Kim, J-. & Lee, J-. Multiple-route recombination dynamics and improved stability of perovskite quantum dots by plasmonic photonic crystal., *J. Phys. Chem. Lett.* 2022, **13**, 5040-5048.

(Supplementary; Supplementary Fig. 14, Fig.15 and Supplementary Table S4 were added)

Supplementary Fig. 14. TRPL spectroscopy of R-NEA and controlled R-NEA OIHP thin films measured with excitation wavelength of 371nm. The traces fitted to a biexponential decay function of $y(t) = A_1 \exp(-t/\tau_1) + A_2 \exp(-t/\tau_2)$ with adj. $R^2 > 0.99$.

Table S4. Fitting parameters of TRPL decay of NEA isomer OIHP thin films.

Commented [손1]: Nuzzo, D. *et al.* Circularly polarized photoluminescence from chiral perovskite thin films at room temperature. *ACS Nano* 2020, **14**, 7610-7616 논문은 본문 ref 7 이라 따로 추가하지 않았습니다.

	A1 (%)	τ_1	A2 (%)	τ_2	Adj. R ²
R-1NEA	97.08	0.280	2.92	2.068	0.992
Controlled R-1NEA	97.86	0.289	2.14	2.794	0.99
R-2NEA	65.60	0.519	34.40	1.985	0.996
Controlled R-2NEA	62.66	0.47	37.34	1.857	0.997

Supplementary Fig. 15. Schematic illustration of energy diagram perturbation by chirality and the CPPL emission mechanism. The conduction band minimum (CBM) and valence band maximum (VBM) are composed of the states with total angular momentum quantum number ($J = \frac{1}{2}$) and magnetic quantum number (m_s) of $\pm\frac{1}{2}$ according to the spin-state of electrons. The σ^+_{abs} and σ^-_{abs} indicate the absorption of the RCP (+) and LCP(-), respectively. The σ^+_{abs} corresponds to the excitonic transition where the magnetic quantum number changes from $-\frac{1}{2}$ to $+\frac{1}{2}$ (vice versa for σ^-_{abs}). The excitonic states can be perturbed when the chirality is transferred to the OIHP lattice. In this case, the energy states with up-spin state ($m_s = +\frac{1}{2}$) and down-spin state ($m_s = -\frac{1}{2}$) are no longer identical. Therefore, the photon emitted from the chiral OIHPs can be spin-polarized.

Commented [문2]: 그림하고 캡션 간격 여러 군데에서 스페이싱 두었음. 정확히 반영할 것

Commented [문3R2]: 정렬도 여기서 양쪽 정렬 사용했음

본문에서는 원안대로 (왼쪽이면 그렇게)

Comment 3:

There is no any proper conclusions in the manuscript and no any outlook on the potential applications of the research. Without that the manuscript does not look complete.

Author's Response:

We appreciate the reviewer's critical comment of future application. The core object of this paper is to scrutinize the effect of different structural isomer configurations on chiroptical response in chiral RP OIHPs and to present the material design rule for developing new-type of semiconductor with excellent intrinsic chirality and stability. However, as reviewer suggested, it is also important to explore the potential application of this research. Therefore, as a proof-of-concept, planar-type circularly polarized light-photodetectors (CPL-PDs) based on chiral RP OIHPs was additionally demonstrated. Fig. R14 describes the structure of CPL-PDs as well as experimental procedures to investigate its discriminating capability between LCP and RCP illumination. Fig. R15 represents the photocurrent vs. time curve of CPL-PDs with different isomer under LCP and RCP illumination by using laser at 365nm, 385 nm, and 400 nm. Both of CPL-PDs exhibited reliable operational stability and distinguishability upon repeated illumination measurement. Fig. R16 shows the photocurrent vs. voltage curve of CPL-PDs with different isomer under LCP and RCP illumination by using the same laser at the applied external voltage range from -4 V to 4 V. It is worth noting that CPL-PDs based on *R*-2NEA exhibited higher photocurrent response to LCP than RCP (Fig. R17), whereas CPL-PDs based *R*-1NEA showed better response to RCP than LCP over entire voltage range, which is consistent with the observed result of CD spectra at the first extinction band edge. Although the current level of our proof-of-concept CPL-PDs is limited to a few of pA level, we clearly demonstrated that the sign conversion phenomenon in CD spectra can be also recognized as a different preferential photocurrent response of CPL-PDs. We expect that important application and enhanced performance can be developed based on our preliminary observation and proposed material design rule.

Fig. R14. Schematic illustration of proof-of-concept planar type CPL-PDs application. The channel between neighboring electrodes had a length of $70\ \mu\text{m}$. The light was generated by LED and lasers with various wavelengths of 365 nm, 385 nm, and 400 nm. The unpolarized light was converted to circularly polarized light by using linear polarizer and quarter-waveplate (Thorlabs, LPVISA050).

Fig. R15. Photocurrent-time curve under LCP and RCP with the varied CPL wavelength. The photocurrent was measured at an applied voltage of 4V.

Fig. R16. Photocurrent-voltage curve under LCP and RCP light with the varied CPL wavelength. The small boxes show the zoomed-in photocurrent-voltage curve for clarity.

Fig. R17. CPL light source wavelength dependent g_{res} of the CPL-PD. The g_{res} were calculated using the following equation : $g_{res} = \frac{I_L - I_R}{I_L + I_R}$.

Revision made (colored in blue):

(in Page 16-17)

... (as well as more asymmetric hydrogen-bonding interaction) induced by structural isomer with different functional group location.

As a proof-of-concept, planar-type circularly polarized light-photodetectors (CPL-PDs) based on chiral RP OIHPs was demonstrated. Supplementary Fig. 20 describes the structure of CPL-PDs as well as experimental procedures to investigate its capability to discriminate between LCP and RCP illumination. The photocurrent vs time curve of CPL-PDs with different isomers under LCP and RCP illumination by using laser at 365nm, 385 nm, and 400 nm were presented in Supplementary Fig. 21. Both of CPL-PDs exhibited reliable operational stability and distinguishability upon repeated illumination measurement. In addition, the photocurrent vs voltage curve of CPL-PDs with different isomer under LCP and RCP illumination by using the same laser applying external voltage range from -4 V to 4 V were also provided

(Supplementary Fig. 22). It is worth noting that in CPL-PDs based on *R*-2NEA always exhibited higher photocurrent response to LCP than RCP (Supplementary Fig. 23), whereas CPL-PDs based *R*-1NEA showed better response to RCP than LCP over entire voltage range, which is consistent with the observed result of CD spectra at the first extinction band edge. Although the current level of our proof-of-concept CPL-PDs is limited to a few of pA level, we clearly demonstrated that the sign conversion phenomenon in CD spectra can be also recognized as a different preferential photocurrent response of CPL-PDs.

Finally, we carried out a thermal stability test to compare the influence of the chiral isomer cation on the structural integrity of chiral RP OIHPs. Both chiral RP OIHPs with chiral NEA isomer cations showed excellent stability under ambient atmosphere after six weeks, ...

(in Discussion)

... Consequently, 2NEA chiral RP OIHP exhibited more enhanced spin-polarized photon absorption behavior (~40% increase in anisotropy factor) than 1NEA chiral RP OIHP. Furthermore, we also demonstrated that the sign conversion phenomenon in CD spectra can be converted in a different preferential photocurrent response of CPL-PDs. Moreover, 2NEA chiral RP OIHPs also demonstrated exceptional environmental stability under harsh conditions (75 °C and 75% RH) for seven days. Our results suggest that small conformational changes in isomers (such as different locations of functional groups) can lead to greatly differing spin-related and structural properties of chiral RP OIHPs. Our proposed strategy based on structural isomer chiral cations clarifies the relationship between hydrogen-bonding interactions and chiroptical phenomena in chiral RP OIHPs. ~~Eventually, it could provide a pathway toward establishing the material design rules for developing novel chiral 2D OIHPs with strong intrinsic chirality and stability, which is essential for chiro-optoelectronic applications such as CPL detectors or emitters.~~ We expect that important applications and next-generation energy conversion devices with excellent performance may be developed based on our preliminary observations and proposed material design rule.

(Supplementary; Supplementary Fig. 20 ~ Fig.23 were added)

Supplementary Fig. 20. Schematic illustration of proof-of-concept planar type CPL-PDs application. The channel between neighboring electrodes had a length of 70 μm . The light was generated by LEDs and laser with various wavelengths of 365 nm, 385 nm, and 400 nm. The unpolarized light was converted to circularly polarized light by using linear polarizer and quarter-waveplate (Thorlabs, LPVISA050).

Commented [문4]: LED laser 아닌가? and 불필요?
해당 다른 부분도 수정

Commented [손5R4]: 연구실 내 프로브스테이션에 365nm, 385nm light source 는 LED 이고 400nm light source 는 레이저로 구성되어있어서 and 를 추가했습니다.

Supplementary Fig. 21. Photocurrent-time curve under LCP and RCP with the varied CPL wavelength. The photocurrent was measured at an applied voltage of 4V.

Supplementary Fig. 22. Photocurrent-voltage curve under LCP and RCP light with the varied CPL wavelength. The small boxes show the zoomed-in photocurrent-voltage curve for clarity.

- Commented [문6]:** 그림내 점선 화살표를 넣어서 (왼쪽으로) 크게 보여주는 영역 표시
- Commented [손7R6]:** 수정하였습니다.
- Commented [문8R6]:** 위하고 동일하게 반영해야. 여긴 누락

Supplementary Fig.23. CPL light source wavelength dependent g_{res} of the CPL-PD. The

g_{res} were calculated using the following equation : $g_{res} = \frac{I_L - I_R}{I_L + I_R}$.

*(experimental; relevant experimental sections were added)
(in Page 23)*

Device fabrication of CPL-PDs and characterization

The soda lime glass substrate was cleaned as above mentioned. After UV-ozone treatment, the 20wt% precursor was spin-coated onto the soda lime glass at 2000 rpm for 30s before annealing for 30min at 120°C. After the annealing, 70nm thick Au top electrode was deposited onto the thin films with thermal evaporation. The CPL photocurrent vs time curves and photocurrent vs voltage curves of PDs were measured at a bias voltage of 4V using an Agilent 4155C in a probe station.

REVIEWERS' COMMENTS

Reviewer #1 (Remarks to the Author):

The revised manuscript was highly improved after revision. Authors replied to the all questions raised by the reviewers, and the answers were almost in satisfactory level with substantial supplements of new data from extra experiments. I recommend this article to be published as it is.

Reviewer #2 (Remarks to the Author):

The author has made some improvements and may be able to be published

Reviewer #3 (Remarks to the Author):

The authors have done great job in revising their manuscript. All issues have been addressed. Therefore the manuscript can be accepted in present revised form.

Response Letter

<Reviewer 1>

The revised manuscript was highly improved after revision. Authors replied to the all questions raised by the reviewers, and the answers were almost in satisfactory level with substantial supplements of new data from extra experiments. I recommend this article to be published as it is.

Author's Response:

We appreciate the reviewer for affirmative comment on our revised manuscripts.

<Reviewer 2>

The author has made some improvements and may be able to be published

Author's Response:

We would like to thank the reviewers for providing valuable feedback to improve our work.

<Reviewer 3>

The authors have done great job in revising their manuscript. All issues have been addressed. Therefore the manuscript can be accepted in present revised form.

Author's Response:

We are grateful to the reviewers for the constructive comments on our manuscript.

Revision made (colored in blue):

We have thoroughly checked the entire manuscript and have corrected some minor mistakes. Also, according to the policy of *Nature Communications*, we change "Supplementary Fig. X" to "Supplementary Figure X".

(Supplementary Information Page 1)

This PDF file includes:

Supplementary ~~Fig~~ Figure.1 to ~~15~~ 26

Supplementary Note 1

Supplementary Table 1 to ~~3~~ 4

References

Revision made (colored in blue):

The arrangement of Fig. 1 was changed to make it easier for readers to clearly identify the hydrogen bonding sites of Fig.1a and b.

(The arrangement of Fig. 1 was changed)

Fig. 1. Crystal structure of chiral isomer RP OIHPs and the hydrogen bonding interactions between the chiral organic spacer and the inorganic framework.

Revision made (colored in blue):

According to the editor's checklist, we correct some texts of the manuscript. We are thankful for her/his delicate guidance.

(in page 1)

In principle, the induced chirality of hybrid ~~perovskite~~-perovskites results from symmetry-breaking within inorganic frameworks. However, the detailed mechanism behind the chirality transfer remains unknown due to the lack of systematic studies. Here, using the structural isomer with different functional group location, we ~~can~~ deduce the effect of hydrogen-bonding interaction between two building blocks on the degree of chirality transfer in inorganic frameworks. To the best of our knowledge, the effect of asymmetric hydrogen-bonding interaction on chirality transfer was firstly verified through experimental demonstration. Systematic studies of crystallography parameters confirm that the different asymmetric hydrogen-bonding interactions derived from different functional group location play a key role in chirality transfer phenomena and the resulting spin-related properties of chiral perovskites. The methodology to control the asymmetry of hydrogen-bonding interaction through the small structural difference of structure isomer cation can provide rational design paradigm for unprecedented spin-related properties of chiral perovskite.

(in page 21)

~~Experimental Section/Methods~~ **Methods**